# Potent latency reversal by Tat RNA-containing nanoparticle enables multi-omic analysis of the HIV-1 reservoir

Marion Pardons [1,8], Basiel Cole[1,8], Laurens Lambrechts [1,2], Willem van Snippenberg [1], Sofie Rutsaert[1], Ytse Noppe[1], Nele De Langhe[1], Annemieke Dhondt[3], Jerel Vega[4], Filmon Eyassu[5], Erik Nijs[6], Ellen Van Gulck[6], Daniel Boden[7] & Linos Vandekerckhove [1] ✉

The development of latency reversing agents that potently reactivate HIV without inducing global T cell activation would benefit the field of HIV reservoir research and could pave the way to a functional cure. Here, we explore the reactivation capacity of a lipid nanoparticle containing Tat mRNA (Tat-LNP) in CD4 T cells from people living with HIV undergoing antiretroviral therapy (ART). When combined with panobinostat, Tat-LNP induces latency reversal in a significantly higher proportion of latently infected cells compared to PMA/ionomycin (≈ 4-fold higher). We demonstrate that Tat-LNP does not alter the transcriptome of CD4 T cells, enabling the characterization of latently infected cells in their near-native state. Upon latency reversal, we identify transcriptomic differences between infected cells carrying an inducible provirus and non-infected cells (e.g. *LINC02964*, *GZMA*, *CCL5*). We confirm the transcriptomic differences at the protein level and provide evidence that the long non-coding RNA *LINC02964* plays a role in active HIV infection. Furthermore, p24+ cells exhibit heightened PI3K/Akt signaling, along with downregulation of protein translation, suggesting that HIV-infected cells display distinct signatures facilitating their long-term persistence. Tat-LNP represents a valuable research tool for in vitro reservoir studies as it greatly facilitates the in-depth characterization of HIV reservoir cells' transcriptome and proteome profiles.

The long-term persistence of latent HIV in various cellular and anatomical reservoirs in people treated with antiretroviral therapy (ART) is a major hurdle to HIV eradication[1–3]. Strong mitogens, such as phorbol myristate acetate (PMA) and anti-CD3/CD28 antibodies, are the most potent latency-reversing agents (LRAs) available to date, making them the gold standards for in vitro assays that aim to achieve maximal

reactivation[2,4–9]. Nevertheless, these molecules severely alter the phenotype of CD4 T cells[10–12], impairing the analysis of the inducible HIV-1 reservoir in its near-native state. As mitogens are not translatable to the clinic due to their toxicity and lack of specificity[13], drug repurposing studies have identified well-tolerated LRAs for use in humans, such as histone deacetylase inhibitors (HDACi)[14–22] and protein kinase C (PKC)

[1]HIV Cure Research Center, Department of Internal Medicine and Pediatrics, Ghent University Hospital, Ghent University, 9000 Ghent, Belgium. [2]BioBix, Department of Data Analysis and Mathematical Modelling, Faculty of Bioscience Engineering, Ghent University, 9000 Ghent, Belgium. [3]Department of Nephrology, Ghent University Hospital, 9000 Ghent, Belgium. [4]Arcturus Therapeutics, 10628 Science Center Drive, Suite 250, San Diego 92121 CA, USA. [5]Computational biology, Johnson and Johnson, 2340 Beerse, Belgium. [6]Janssen infectious diseases and diagnostics, Johnson and Johnson, 2340 Beerse, Belgium. [7]Janssen Biopharma, Johnson and Johnson, South San Francisco 94080 CA, USA. [8]These authors contributed equally: Marion Pardons, Basiel Cole. ✉e-mail: linos.vandekerckhove@ugent.be

agonists[23,24]. However, these compounds do not reach the levels of reactivation observed with mitogens in vitro[25–31], and their administration to ART-treated individuals does not result in a significant reduction of the HIV reservoir size nor in a delay in the time to rebound upon ART cessation[14–23]. The viral protein Tat, by playing a key role in HIV transcription[32] and regulating the switch between active and latent infection[33], appears to be an ideal candidate for latency reversal approaches. While previous studies have successfully used exogeneous Tat to reactivate HIV from latency in cell lines[34,35] and primary cells from ART-treated individuals[36–38], Tat proteins are known to become trapped in the endosomes, potentially limiting their reactivation potential[39]. To overcome this, a lipid nanoparticle containing a Tat mRNA molecule has recently been developed, called LNP-2-T66 RNA (referred to as Tat-LNP in this manuscript)[38]. Tat-LNP was previously shown to reactivate HIV in primary CD4 T cells from ART-treated individuals[38]. Here, we studied the synergistic effect of Tat-LNP with the histone deacetylase inhibitor panobinostat (PNB) and compared the reactivation potential of this combination to the current gold standard PMA/ionomycin (PMA/i) using the HIV-Flow assay[8]. Furthermore, we investigated overlap between HIV-1 proviruses reactivated with Tat-LNP/PNB and PMA/i using the recently developed STIP-Seq assay[40], which enables to simultaneously retrieve the integration site (IS) and the near full-length (NFL) proviral sequence from single-sorted p24+ cells. Finally, we demonstrated that Tat-LNP does not alter the phenotype of CD4 T cells and highlighted the potential of using Tat-LNP as a tool to uncover transcriptomic and proteomic differences between HIV-infected CD4 T cells carrying an inducible provirus and non-infected CD4 T cells.

## Results

### Tat-LNP in combination with PNB induces HIV latency reversal in a higher fraction of cells than PMA/i

To identify the best timepoint for measuring HIV latency reversal, CD4 T cells from four ART-treated individuals (Supplementary Table 1) were stimulated for 24 h or 48 h with Tat-LNP (250 ng/mL) and PNB (50 nM) individually or combined (Tat-LNP/PNB). A 24h-stimulation with PMA/i (162 nM/1 µg/mL) served as a positive control, as previously defined[8]. Three negative control conditions were included: non-stimulated cells (NS), DMSO, and a lipid nanoparticle containing the hemagglutinin peptide from influenza virus (HA-LNP). Reactivation capacity was assessed at the level of intracellular p24 protein production by the HIV-Flow assay[8] and at the level of viral particle release by ultrasensitive p24-SIMOA (Fig. 1).

Tat-LNP did not induce cell death at the two-time points tested, indicating that this formulation is not cytotoxic (Fig. 1A). No p24+ cells were detected by HIV-Flow in the negative control conditions (NS, DMSO, HA-LNP) (Fig. 1B). PNB induced no or low levels of HIV reactivation, both at 24 h and 48 h post-stimulation (median frequency = 0.3 p24+ cells/$10^6$ CD4 T cells at both time points). In contrast, higher frequencies of p24+ cells were observed in the Tat-LNP condition compared to PMA/i in 2/4 individuals at 24 h post-stimulation (median fold induction = 0.8) and in 4/4 participants at 48 h post-stimulation (median fold induction = 1.8; Fig. 1B, Supplementary Fig. 1A). The highest levels of reactivation were observed at 24 h post-stimulation with the combination of Tat-LNP and PNB (median frequency = 18.1 p24+ cells/$10^6$ CD4 T cells), with a median fold induction of 3.2 compared to PMA/i (Fig. 1B, Supplementary Fig. 1A). Limited cell death was observed with this combination (median viabilities = 73.4% for Tat-LNP/PNB and 84.9% for NS; Fig. 1A). Frequency of p24+ cells in the Tat-LNP/PNB condition markedly dropped at 48 h (median frequency = 9.3 p24+ cells/$10^6$ CD4 T cells; Fig. 1B, Supplementary Fig. 1A), which may be partially explained by the lower cell viability at that timepoint (median viabilities = 53.1% for Tat-LNP/PNB and 77% for NS; Fig. 1A). Assessment of viral particle release in the culture supernatant by p24-SIMOA confirmed the results obtained by HIV-Flow, with the highest

fold inductions relative to PMA/i observed in the Tat-LNP/PNB condition both at 24 and 48 h post-stimulation (median fold inductions of 1.4 at 24 h and 15.2 at 48 h post-stimulation; Fig. 1C).

Because Tat-LNP/PNB displayed the highest levels of reactivation at 24 h post-stimulation, we applied this combination on CD4 T cells from a larger cohort of ART-treated individuals (n = 22, Supplementary Table 1), and compared the levels of reactivation to those obtained with PMA/i. The frequency of p24+ cells was significantly higher in the Tat-LNP/PNB condition compared to the PMA/i treatment (p < 0.00001, median frequencies = 0.9 and 4.1 p24+ cells/$10^6$ CD4 T cells for PMA/i and Tat-LNP/PNB, respectively), with a median fold increase of 3.9 (Fig. 1D). Tat-LNP/PNB stimulation of CD4 T cells from four ART-treated individuals gave reproducible measurements with the HIV-Flow assay (4–6 independent experiments; mean coefficient of variation = 0.21; Supplementary Fig. 1B). Of note, although frequencies of p24+ cells were higher in the Tat-LNP/PNB condition compared to PMA/i, the p24 intracellular production per cell (as assessed by median fluorescence intensities (MFI) for both p24 antibodies) tended to be higher in the PMA/i-stimulated cells (median MFI APC = 1067 and 1762, median MFI FITC = 2837 and 4654 for Tat-LNP/PNB and PMA/i, respectively; Supplementary Fig. 1C).

In conclusion, we show that Tat-LNP can efficiently reactivate HIV from latency with a minimal impact on cell viability. Moreover, the highest frequencies of p24+ cells and virus release were observed when Tat-LNP was combined with PNB, indicating a synergistic effect between these two compounds to induce latency reversal.

### Tat-LNP/PNB-stimulated p24+ cells express low levels of CD4 and are enriched in the effector memory fraction

We first assessed T cell activation in response to Tat-LNP and PNB. Tat-LNP did not induce upregulation of cell activation markers at the cell surface, as evidenced by the stable expression of CD69, CD25 and HLA-DR over time (Supplementary Fig. 2A). In contrast and as previously documented[41,42], PNB induced the upregulation of CD69 compared to the NS condition (mean MFI = 661.3 and 86.0, respectively), while the levels of CD25 and HLA-DR remained unaltered (Supplementary Fig. 2B).

Despite PNB-induced CD69 upregulation, stimulation with Tat-LNP/PNB preserved CD4 expression in bulk CD4 T cells, along with CD4 T cell subsets (Supplementary Fig. 2C). This enabled the assessment of CD4 expression at the surface of p24+ cells following latency reversal: p24+ cells expressed significantly lower levels of CD4 when compared to p24− cells (p = 0.0002, median MFI = 70 and 3426 for p24+ and p24− cells, respectively; Fig. 2A, B), likely as a consequence of the activity of the viral protein Nef[43]. We also analyzed the memory phenotype of p24+ cells following stimulation with Tat-LNP/PNB (Fig. 2C). Tat-LNP/PNB-stimulated p24+ cells were significantly enriched in the effector memory compartment (TEM; p = 0.001; Fig. 2C), as previously described in the context of PMA/i stimulation[8,44]. In contrast, only a low number of p24+ cells displayed a naive (TN) or a terminally differentiated (TTd) phenotype (Fig. 2C). We also compared the phenotype of p24+ cells between the Tat-LNP/PNB condition and the PMA/i treatment (Fig. 2D, Supplementary Fig. 2B). Although no significant differences were observed between the two stimulation conditions for each analyzed subset, there was a trend toward more p24+ cells with a central/transitional memory phenotype (TCM/TTM) in the Tat-LNP/PNB condition compared to the PMA/i condition (6/8 participants, p = 0.148), suggesting that different classes of LRAs might preferentially induce latency reversal in infected cells with distinct phenotypes. Of note, the assessment of CD69 expression in memory subsets revealed that CD69 is expressed at lower levels in the TCM compared to the TEM subset in the absence of stimulation (mean MFI = 99.8 and 131.8, respectively) (Supplementary Fig. 2D). In contrast, following PNB stimulation, we observed greater levels of CD69 expression, along with a greater fold increase relative to NS, within the TCM compared to the TEM subset

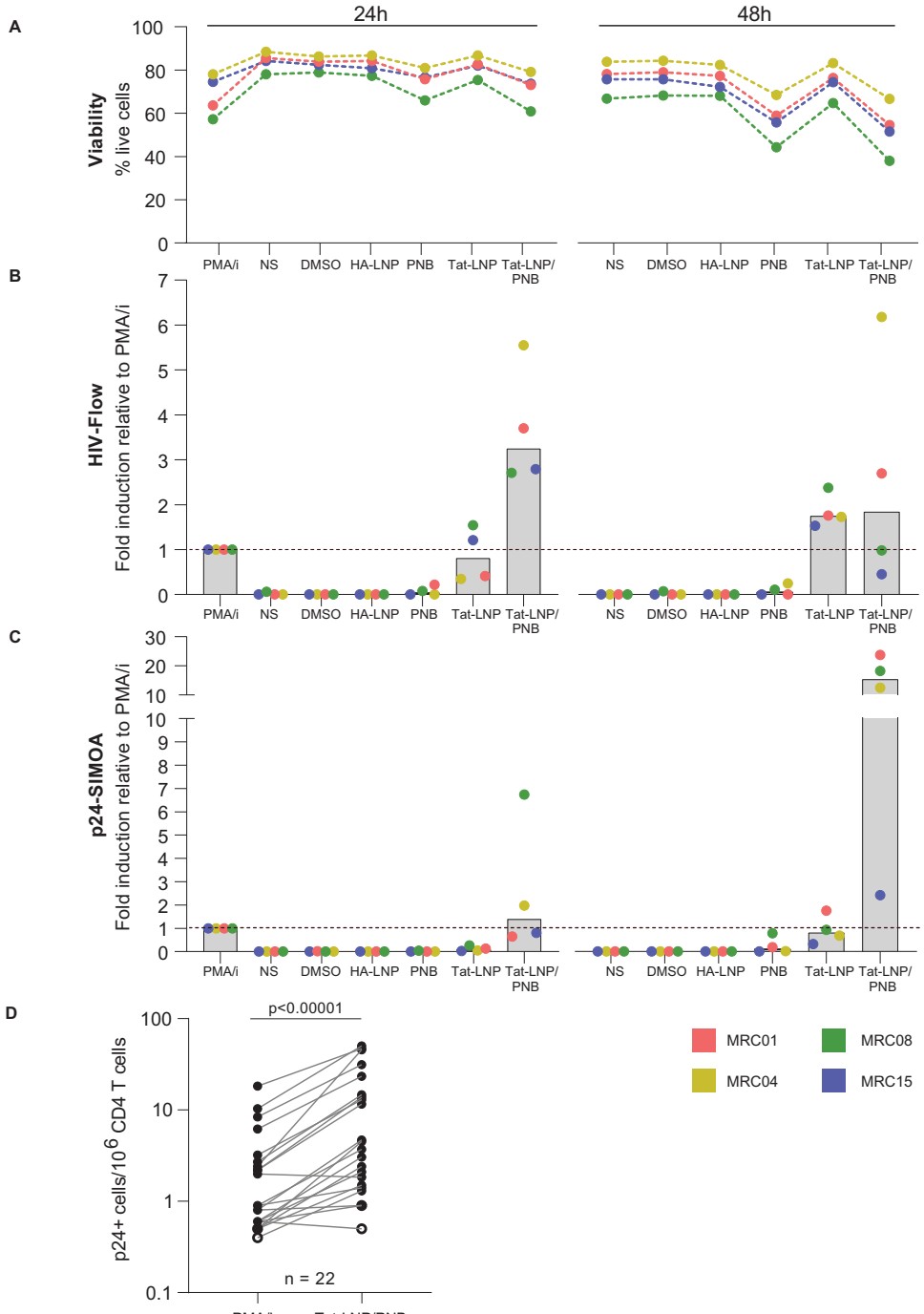

**Fig. 1 | Tat-LNP in combination with PNB induces latency reversal in a higher fraction of cells than PMA/i. A–C** CD4 T cells from *n* = 4 ART-treated individuals were stimulated for 24 h or 48 h with Tat-LNP and PNB alone or combined (Tat-LNP/PNB). A 24h-stimulation with PMA/i was used as a positive control. **A** Percentage of live cells (as defined by a negative Live/Dead stain) among all recorded events. **B** HIV-Flow was used to measure the frequency of p24+ cells following reactivation: data are represented as fold inductions relative to frequencies obtained by PMA/i at 24 h. Gray columns depict median values. **C** p24-SIMOA was used to measure the concentrations of p24 in the culture supernatants following reactivation: data are represented as fold inductions relative to p24 concentrations measured by PMA/i at 24 h. Gray columns depict median values. **D** Frequencies of p24+ cells as measured by HIV-Flow in PMA/i versus Tat-LNP/PNB-stimulated cells (*n* = 22 ART-treated individuals). For statistical analysis, a two-sided non-parametric Wilcoxon test was used (*p* = 0.000004). Source data are provided with this paper.

(mean MFI = 853.5 and 731, mean fold increase = 9.0 and 5.5, respectively) (Supplementary Fig. 2D, E). The increased effectiveness of PNB in promoting CD69 expression within the TCM subset compared to the TEM subset, could offer a partial explanation for the observed tendency toward a greater presence of p24+ cells with a TCM phenotype following Tat-LNP/PNB stimulation.

## HIV-1 proviruses reactivated by Tat-LNP/PNB mostly overlap with those retrieved by PMA/i

We took advantage of the recently developed STIP-Seq assay[40], which allows for the simultaneous assessment of the IS and the NFL proviral sequence from single-sorted p24+ cells, to compare HIV-1 proviruses reactivated with Tat-LNP/PNB and PMA/i (Supplementary Fig. 3,

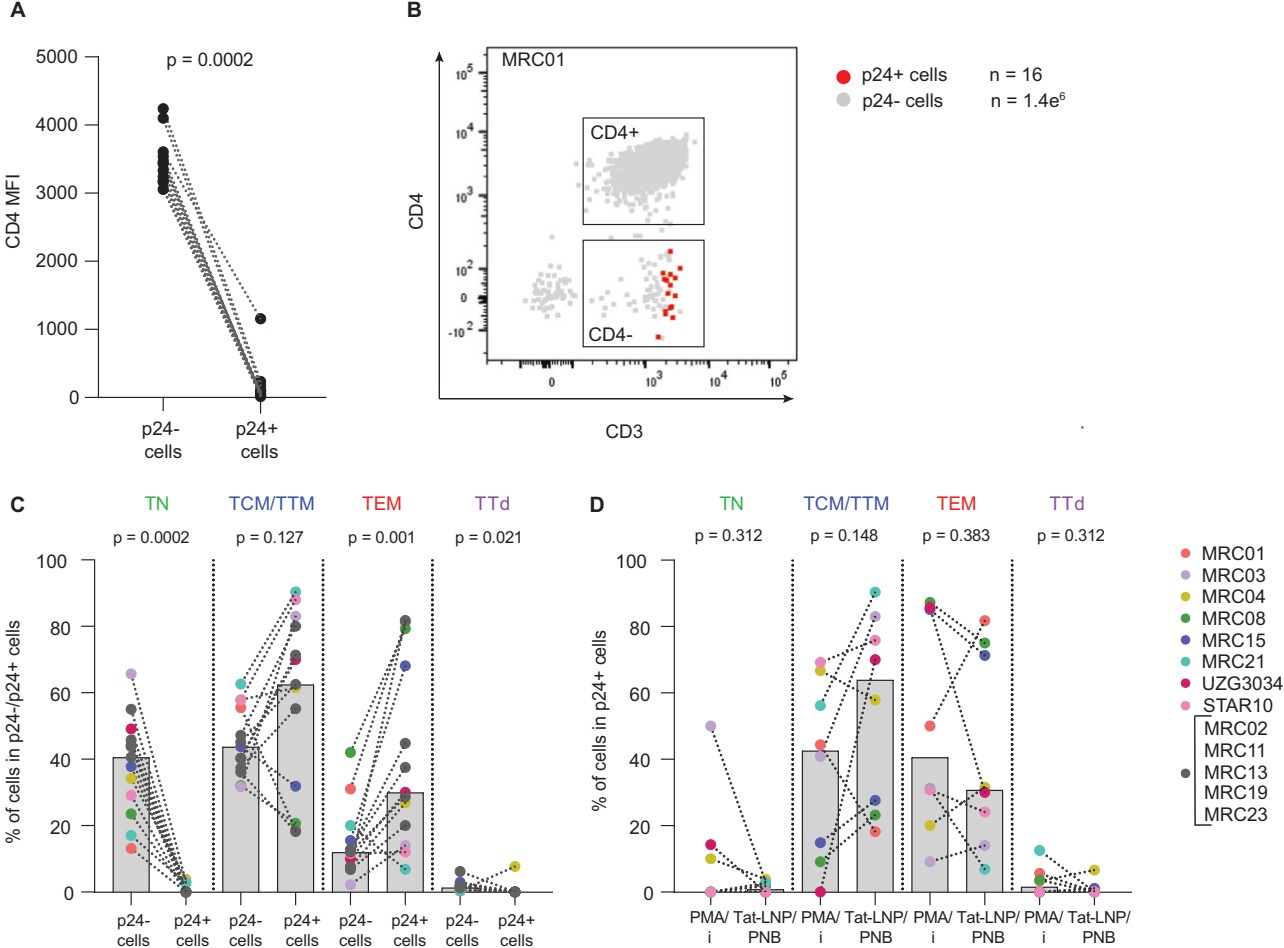

**Fig. 2 | Tat-LNP/PNB-stimulated p24+ cells express low levels of CD4 and are enriched in the effector memory fraction. A–D** CD4 T cells were stimulated for 24 h with Tat-LNP/PNB and PMA/i. Participants with a minimal number of 7 p24+ cells are represented. **A** Median fluorescence intensity (MFI) for CD4 expression is compared between p24− cells and p24+ cells (*n* = 13 ART-treated individuals); gated on LiveDead−/CD8−/CD3+/CD4− and CD4+ cells. **B** Representative dot plot showing the CD4 phenotype of p24+ cells (in red) overlaid to p24− cells (in gray). **C** Percentage of cells with a given phenotype (TN, TCM/TTM, TEM, TTd) in the p24+ and p24− fractions (*n* = 13 ART-treated individuals). HIV-Flow data (PFA-fixed cells)

are used to generate Fig. 2C. **D** Percentage of cells with a given phenotype (TN, TCM/TTM, TEM, TTd) in the p24+ fraction following stimulation with PMA/i or Tat-LNP/PNB (*n* = 8 ART-treated individuals). HIV-Flow (PFA-fixed cells) and STIP-Seq (methanol-fixed cells) phenotypic data are combined to generate Fig. 2D. Gray bars depict median values. For statistical analyses, two-sided non-parametric Wilcoxon tests were used. TN naive T cells, TCM/TTM central and transitional memory T cells, TEM effector memory T cells, TTd terminally differentiated T cells. Source data are provided with this paper.

Supplementary Data 1). A total of 173 NFL sequences were obtained from the 5 participants analyzed.

As previously shown[40,45,46], we observed that a large proportion of the translation-competent proviruses displayed packaging signal (PSI) and/or major splice donor (MSD) defects (157/173, 90.8%), while only a minority were genome-intact (16/173, 9.2%) (Fig. 3A, Supplementary Data 1). Other defects, such as inversions, large internal deletions, hypermutations and frameshifts, were not observed. Moreover, the proportions of genome-intact or PSI/MSD-defective proviruses did not differ significantly between PMA/i (*n* = 64) or Tat-LNP/PNB-stimulated conditions (*n* = 109) (intact: 9.4% and 9.2%, PSI/MSD-defective: 90.6% and 90.8% for PMA/i and Tat-LNP/PNB, respectively; Fig. 3A, Supplementary Data 1).

IS analysis showed that most of the clones (as defined by recurrent identical IS or NFL sequences observed through our study and the study from Cole et al.[40]) retrieved with PMA/i were also retrieved with Tat-LNP/PNB (4/5 in MRC01, 4/5 in MRC04, 3/4 in MRC08, 1/1 in MRC15, 1/2 in STAR10) (Fig. 3). Of note, some infected clonal populations seemed to be over-represented in one of the two conditions (e.g. *MLLT3* and *ERGIC2* in MRC01, *SCML4* in MRC08), suggesting that different LRAs might favor the preferential reactivation of specific HIV-

infected clones over others. Besides, some minor clones that were not detected in the PMA/i condition were retrieved in Tat-LNP/PNB-stimulated cells (+5 in MRC04, +1 in MRC08, +1 in MRC15, +3 in STAR10) (Fig. 3B).

## Single-cell RNA-sequencing of p24+ cells allows to study viral transcripts and splice sites

We slightly modified the STIP-Seq assay[40], which relies on methanol fixation and permeabilization, to make it compatible with downstream single-cell RNA-sequencing (scRNA-seq) through Smart-seq2 (see Methods)[47,48]. According to the most optimal reactivation conditions determined earlier (Fig. 1B), CD4 T cells from ART-treated individuals were stimulated for 24 h with Tat-LNP/PNB (*n* = 7) and PMA/i (*n* = 2), and for 48 h with Tat-LNP alone (*n* = 5), followed by single-cell sorting and Smart-seq2 processing. This yielded a total of 108 and 212 p24+ cells, as well as 109 and 150 p24− cells following treatment with Tat-LNP alone or combined with PNB, respectively (Supplementary Data 2). In addition, 28 PMA/i-stimulated p24+ cells and 17 NS cells were sorted as positive and negative controls. Total HIV DNA measurements in bulk CD4 T cells (as assessed by digital PCR), and frequencies of p24+ cells following treatment with Tat-LNP alone or

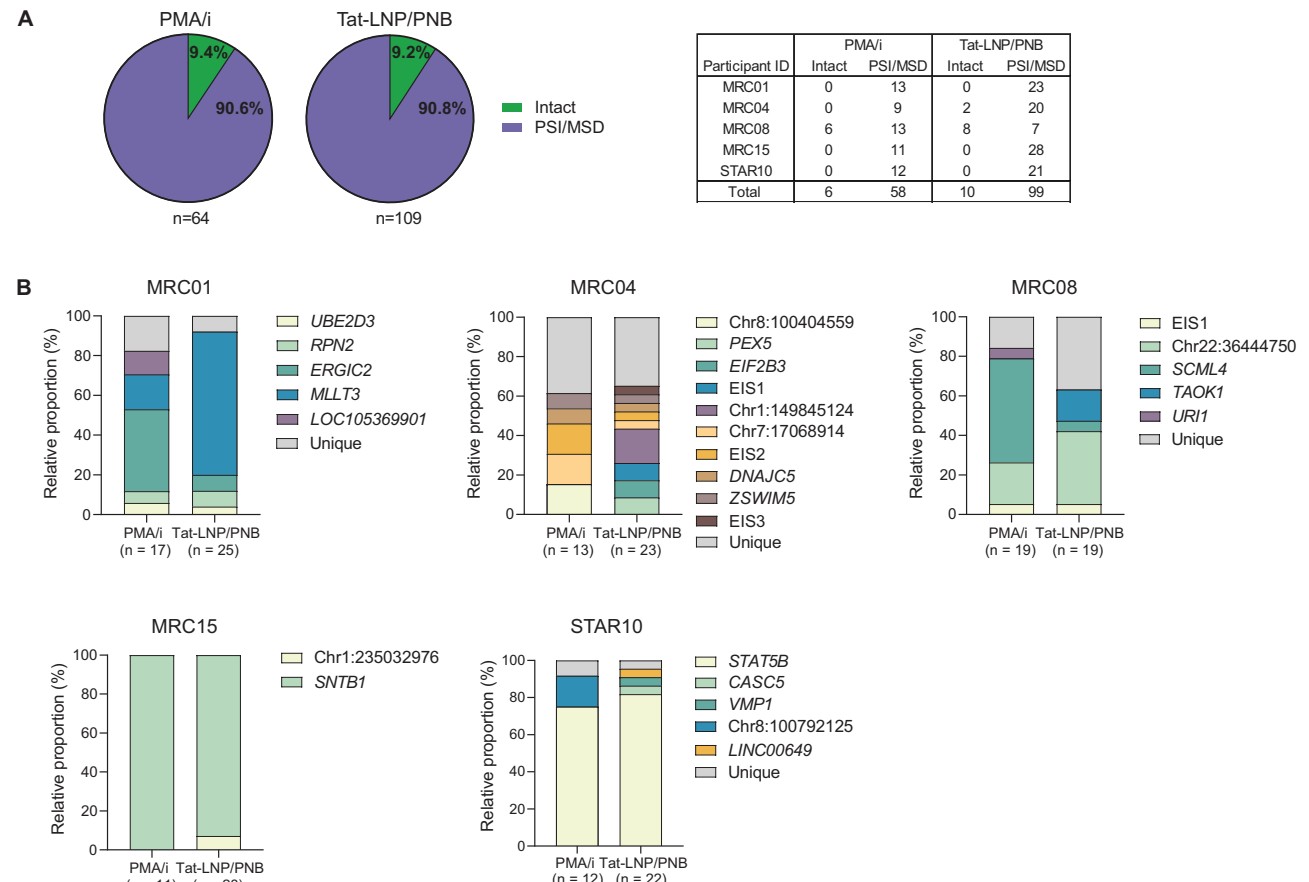

**Fig. 3 | HIV-1 proviruses reactivated by Tat-LNP/PNB mostly overlap with those retrieved by PMA/i. A, B** CD4 T cells from 5 ART-treated individuals were stimulated for 24 h with Tat-LNP/PNB and PMA/i. P24+ cells were single-cell sorted and subjected to STIP-Seq (median number of p24+ cells per participant = 23 for Tat-LNP/PNB and 13 for PMA/i). **A** Pie charts showing the fraction of sorted p24+ cells carrying an intact or a PSI/MSD-defective provirus. The counts of intact and PSI/MSD-defective proviruses for each participant and for each stimulation condition are reported in the accompanying table. **B** Bar plots comparing the relative proportion of each proviral population between PMA/i and Tat-LNP/PNB-stimulated cells. EIS expansion of identical sequences (several proviruses with the same near full-length sequence were retrieved but the IS could not be identified). Supplementary Data 1 provides information on the single-sorted p24+ cells for STIP-Seq analyses.

combined with PNB (as assessed by the HIV-Flow assay) are represented in Supplementary Table 2.

As expected, we did not detect HIV transcripts in the p24– cells following stimulation with Tat-LNP or Tat-LNP/PNB (Supplementary Fig. 4A). Among p24+ cells, the proportion of reads mapping to the reference HXB2 was higher in the Tat-LNP/PNB condition compared to Tat-LNP-stimulated cells ($p = 7.4e^{-9}$, medians = 15.0% for Tat-LNP/PNB and 6.7% for Tat-LNP alone; Supplementary Fig. 4A), indicating that the combination of Tat-LNP with PNB induces higher levels of HIV transcription per-cell compared to Tat-LNP alone. Interestingly, when analyzing all p24+ cells obtained in the three treatment conditions (Tat-LNP, Tat-LNP/PNB, PMA/i), the proportion of HIV reads was significantly higher in the TEM subset compared to the TCM/TTM fraction ($p = 0.04$; Supplementary Fig. 4B), which is in line with a higher degree of DNA accessibility in more differentiated subsets[49]. Levels of HIV transcription (as defined by the normalized percentage of reads mapping to HIV) also positively correlated with the intracellular p24 protein production per cell (as defined by the normalized MFI for p24 antibodies) ($p = 5.1e^{-12}$ for p24 APC and $p = 1.5e^{-10}$ for p24 FITC; Supplementary Fig. 4C).

Following de novo assembly of viral reads, complete coverage of the viral genome was obtained in 46.5% of the p24+ cells that contained HIV transcripts (Supplementary Data 2). By constructing a maximum-likelihood phylogenetic tree containing only completely covered viral sequences from STIP-Seq and Smart-seq2, we showed that identical viral sequences could be retrieved with both techniques (Fig. 4A). Of note, the Smart-seq2 approach identified viruses that could not have been fully retrieved by STIP-Seq due to the deletion or mutation of the 5′ primer binding site (e.g. MRC01, provirus with IS in *RPN2*). Additionally, the fact that scRNA-seq relies on poly-A tail capture rather than on HIV-specific primers presents an additional advantage over STIP-Seq, namely that it can be used to retrieve viral sequences of any subtype (e.g. MRC03, subtype F1, Supplementary Table 1). Starting from the principle that proviruses that are identical over the entire length of the genome most likely originate from the same clonal infected cell population, we inferred the IS of Smart-seq2 viral sequences based on the IS retrieved by STIP-Seq and showed that infected clonal cell populations were retrieved in relatively the same proportions by the two assays following Tat-LNP/PNB stimulation, indicating that there is no sampling bias between the two methods (Fig. 4B).

By analyzing donor splice sites in the 5′ UTR region, we showed that inducible proviruses with defective MSD (D1)/cryptic donor (CD; D1c) sites can use alternative donor splice sites that are located up- or downstream the MSD/CD (Fig. 4C). This observation suggests that MSD-defective proviruses can readily produce spliced transcripts, which can be translated into viral proteins such as Nef, allowing for the observed downregulation of CD4 by p24+ cells (Fig. 2A, B).

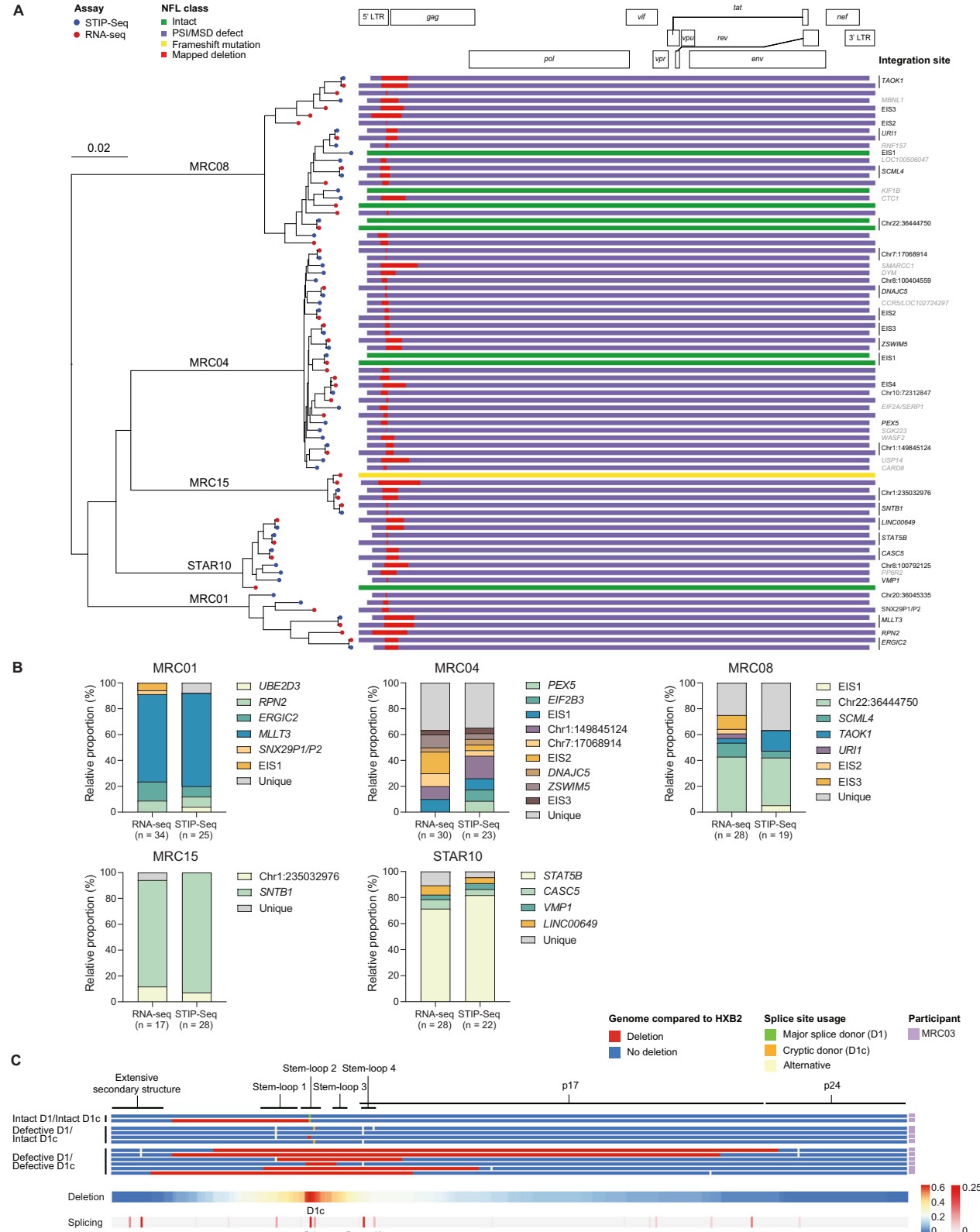

## Tat-LNP does not modify the transcriptome of CD4 T cells

To assess the impact of Tat-LNP individually or in combination with PNB on the cellular transcriptome, bulk CD4 T cells from four ART-treated individuals were stimulated with Tat-LNP and Tat-LNP/PNB as well as their respective controls (NS, DMSO, HA-LNP and PMA/i). In order to capture the immediate cellular responses and initial signaling events to the compounds of interest, cells were stimulated for a period of 6 h. Micro-array gene expression analysis followed by dimensionality reduction and unsupervised clustering revealed three distinct clusters; cluster 1: NS, DMSO, HA-LNP, and Tat-LNP-stimulated cells, cluster 2: PNB and Tat-LNP/PNB-stimulated cells, cluster 3: PMA/i-stimulated cells (Supplementary Fig. 5A). These results show that Tat-LNP alone has a minimal impact on the transcriptome of CD4 T cells. Cluster 2 (PNB and Tat-LNP/PNB) modulated the expression of 6018

**Fig. 4 | Single-cell RNA-sequencing of p24+ cells identifies the same viruses as STIP-Seq. A** Maximum-likelihood phylogenetic tree representing viral sequences for which we have full coverage both with Smart-seq2 (red dots) and STIP-Seq (blue dots). The phylogenetic tree includes sequences from p24+ cells retrieved following Tat-LNP, Tat-LNP/PNB, and PMA/i stimulation. The viral sequences and the corresponding IS are represented on the right-hand side of the phylogenetic tree. Sequences belonging to a clone are represented only once. IS from clonal cells are depicted in black, while IS retrieved only once are depicted in gray. HXB2 is used as a reference genome, and the scale indicates the number of nucleotide substitutions per site. **B** The relative proportions of each clone retrieved by Smart-seq2 and STIP-Seq following Tat-LNP/PNB stimulation are compared. For Smart-seq2, the IS associated to each viral sequence were inferred based on the IS retrieved by STIP-

Seq. EIS expansion of identical sequences (several proviruses with the same near full-length sequence were retrieved but the IS could not be identified). **C** Top panel: virogram depicting the 5' UTR region of representative viral sequences retrieved with Smart-seq2 in MRC03. Viral sequences are grouped into three categories: intact major splice donor (MSD; D1) and cryptic donor (CD; D1c) sites, defective MSD/intact CD, defective MSD and CD. Type of defects and splice site usage are color-coded. Bottom panel: heatmap of the deletions and donor splice sites detected over the entire dataset ($n = 7$ ART-treated individuals; $n = 29$ intact MSD/CD, 12 defective MSD/intact CD, 61 defective MSD/CD sequences). HXB2 is used as a reference genome. Supplementary Data 1 and 2 provide information on the single-sorted p24+ cells for STIP-Seq and Smart-seq2 analyses, respectively.

genes when compared to cluster 1 (adjusted $p$ value $< 0.01$), with upregulated genes mostly involved in cell cycle and ribosome-related pathways (Supplementary Fig. 5B, Supplementary Data 3). Cluster 3 (PMA/i treatment) modified the expression of 6661 genes when compared to cluster 1 (adjusted $p$ value $< 0.01$), with upregulated genes being associated with ribosome-related pathways, response to cytokines, regulation of leukocyte differentiation and T cell activation (Supplementary Fig. 5B, Supplementary Data 3). Prolonged exposure of the cells (24 h) to the compounds of interest produced similar results (Supplementary Data 4).

**P24+ cells display a distinct transcriptional landscape compared to p24− cells**
As Tat-LNP does not significantly modify the transcriptome of CD4 T cells, we took advantage of the scRNA-seq approach to compare the transcriptional landscape of CD4 T cells carrying a translation-competent provirus to non-infected CD4 T cells. Since most of the p24+ cells display a CD45RO+ memory phenotype (Fig. 2C), we used memory p24− cells isolated from the same participants as controls to avoid biases in our analyses. Of note, to limit technical variations, p24− and p24+ cell sorting, Smart-seq2 processing, library preparation and sequencing were conducted simultaneously for each participant.

In line with the micro-array data on bulk CD4 T cells (Supplementary Fig. 5A), sorted cells formed three distinct clusters based on stimulation (Fig. 5A). Tat-LNP-stimulated cells clustered with the NS cells, further supporting that Tat-LNP does not modify the transcriptome of CD4 T cells, while PMA/i and Tat-LNP/PNB-treated cells each formed a separate cluster. The median number of genes detected per cell was the highest in the PMA/i condition, followed by Tat-LNP/PNB-stimulated cells (medians = 1537≈1514 < 1821 < 2306 genes per cell for NS, Tat-LNP, Tat-LNP/PNB, PMA/i, respectively; Supplementary Fig. 6A), which can likely be explained by the fact that Tat-LNP/PNB and PMA/i highly modify the expression of many genes in CD4 T cells (Fig. 5A). In the Tat-LNP/PNB cluster, p24+ and p24− cells clustered together (Fig. 5B) and only 10 transcripts were significantly differentially expressed between these two groups, with the most significant hit being the long non-coding RNA (lncRNA) *LINC02964* (adjusted $p < 0.05$; Supplementary Data 5). As the substantial impact of PNB on the transcriptome of the cells potentially obscured the differences existing between the p24+ and the p24− cells, we focused our differential gene expression (DGE) analysis on p24+ and p24− cells obtained following stimulation with Tat-LNP alone. Unsupervised clustering partially segregated the cells into two groups corresponding to p24− and p24+ cells (Supplementary Fig. 6B). Of note, the imperfect segregation between p24+ and p24− cells further reflects the high similarity between these two cell populations, with only a limited set of DEG observed. We identified 82 genes for which the levels of expression were significantly different between p24+ and p24− cells (adjusted $p < 0.05$; Supplementary Data 5). Among the most differentially expressed genes, we identified upregulation of *LINC02964, SOD1P3, CCL5,* and *GZMA* in p24+ cells, while *ATG10* and *IL7R* were downregulated in p24+ cells compared to p24− (Fig. 5C). To our knowledge,

*LINC02964* and *SOD1P3* have not been reported previously as preferentially expressed by HIV-1 reservoir cells carrying an inducible provirus. Cytotoxic T cell genes such as *GZMA* and *CCL5* were recently shown to be upregulated in HIV-infected cells both in viremic and ART-suppressed individuals[50,51] (Supplementary Table 3). Moreover, *ATG10* and the axis *IL7/IL7R*, by playing central roles in cell survival and cell death, might favor the long-term persistence of HIV reservoir cells.

To identify gene sets that were differentially expressed between p24+ and p24− cells following Tat-LNP stimulation, we conducted gene set enrichment analysis (GSEA). We found that Phosphoinositide 3-kinase (PI3K) signaling was upregulated in p24+ cells, while gene sets associated with protein translation (amino acids metabolism, ribosome biogenesis and assembly, initiation and elongation of translation) were downregulated in p24+ cells compared to p24− cells (Fig. 6A, Supplementary Data 6). Importantly, both pathways were consistently up/downregulated between p24+ and p24− cells across all five participants (Fig. 6B, C), highlighting their potential relevance in HIV infection. The heightened PI3K/Akt signaling, along with the downregulation of protein translation in p24+ cells, might contribute to the long-term persistence of HIV-infected cells by favoring cell survival and promoting latency.

**Transcriptomic differences between p24+ and p24− cells are confirmed at the protein level**
We next sought to investigate whether the transcriptomic hits could be validated at the protein level using flow cytometry. CD4 T cells from 7 ART-treated individuals were stimulated for 48 h with Tat-LNP and protein expression levels of GZMA, GZMB, IL7R and CCL5 were assessed in the p24+ and p24− fractions using the HIV-Flow assay[8]. The expression levels of the proteins of interest (CD45RO, CD27, GZMA, GZMB, IL7R, CCL5) were not modified by Tat-LNP stimulation (Supplementary Fig. 7A). The percentage of CD4 T cells expressing GZMA and GZMB increased progressively with their differentiation stage, with TN cells showing minimal expression and TTd cells exhibiting the highest levels of expression (medians = 1%, 5%, 35%, 61% GZMA+ cells, and 0.1%, 0.1%, 19% and 52% GZMB+ cells, in TN, TCM/TTM, TEM, TTd, respectively; Supplementary Fig. 7B, C). Of note, the fraction of TEM cells expressing GZMA/GZMB was highly variable between the 7 ART-treated participants (range = 13-72% for GZMA and 0.1-61% for GZMB), suggesting that factors other than the phenotype can influence the levels of granzyme expression.

Among CD4 T cells, a higher fraction of p24+ cells expressed GZMA compared to p24− cells ($p = 0.02$; Fig. 7A, Supplementary Fig. 7D). The same trend was observed when the analysis was repeated on the TCM/TTM and TEM subsets separately ($p = 0.03$ for TCM/TTM and $p = 0.08$ for TEM; Fig. 7A). In contrast, a lower fraction of p24+ cells expressed GZMB compared to p24− cells, both in the TCM/TTM and TEM subsets ($p = 0.02$ for TCM/TTM and TEM; Fig. 7B, Supplementary Fig. 7D). Finally, IL7R expression was significantly lower in the p24+ compared to p24− cells among all subsets analyzed ($p = 0.02$ for CD4 T cells, TCM/TTM, and TEM; Fig. 7C, Supplementary Fig. 7D), while CCL5 expression was increased in the p24+ fraction compared to p24−

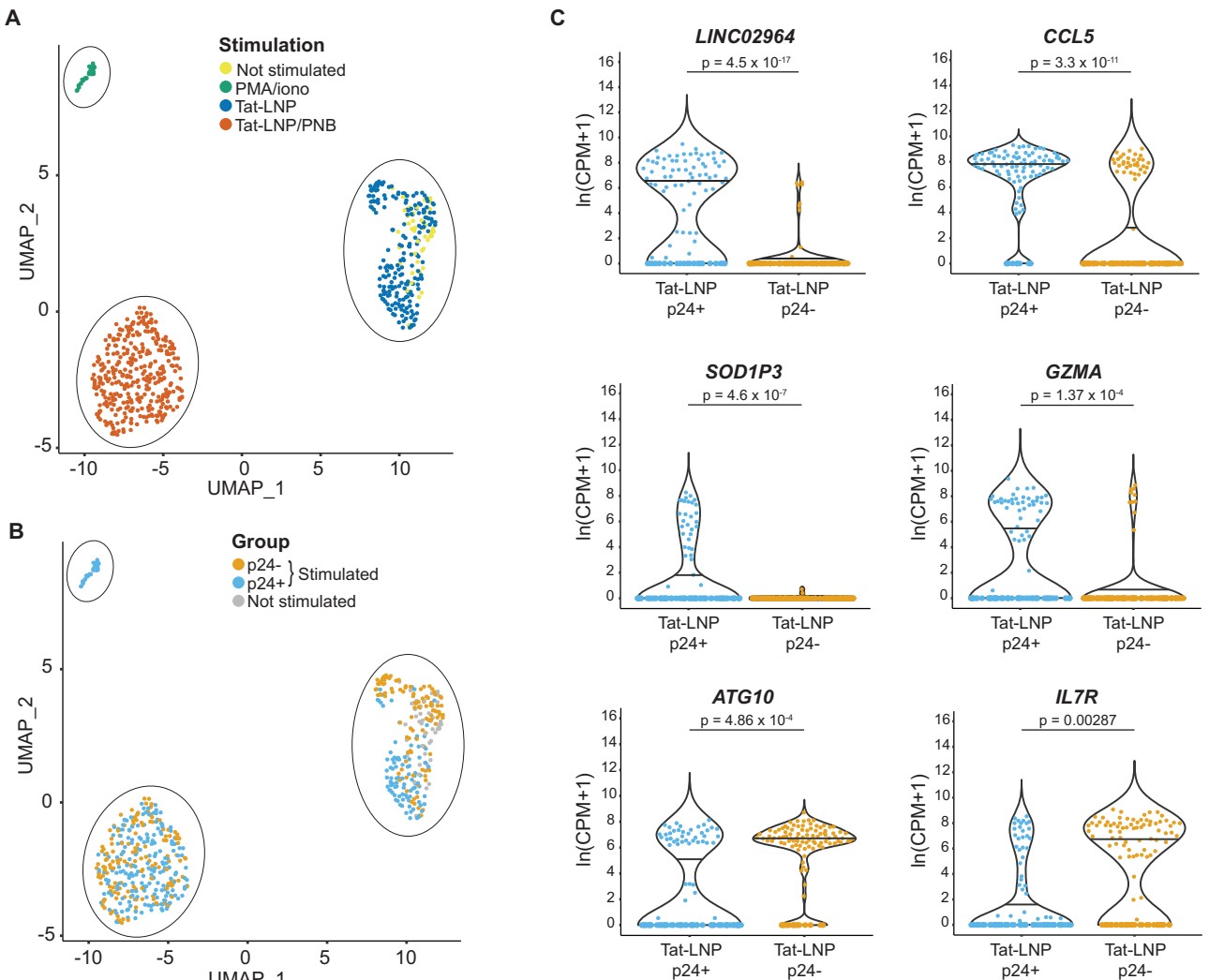

**Fig. 5 | P24+ cells display a distinct transcriptional landscape compared to p24− cells. A**, **B** CD4 T cells were stimulated for 24 h with Tat-LNP/PNB and PMA/i, and for 48 h with Tat-LNP. Uniform manifold approximation and projection (UMAP) of Smart-seq2 data, colored by stimulation (**A**), or p24 expression (**B**). **C** Significantly differentially expressed genes (DEG) between p24+ and p24− cells following a 48h-stimulation with Tat-LNP. Horizontal bars depict median values. *P* values were derived from likelihood ratio tests, with a Bonferroni correction for multiple comparisons. CPM Counts per million mapped reads. The list of DEG between p24+ and p24− cells is shown in Supplementary Data 5.

cells both in total CD4 T cells and the TCM/TTM fraction ($p = 0.03$ and $p = 0.02$, respectively; Fig. 7D, Supplementary Fig. 7D). In conclusion, our findings show that GZMA, IL7R and CCL5 are differentially expressed between p24− and p24+ cells at the protein level, confirming our transcriptomic data.

### Assessment of the role of *LINC02964* in HIV-1 infection and latency reversal

To confirm the potential role of *LINC02964* in HIV-1 pathogenesis, we compared the expression of this transcript in non-infected CD4 T cells and in CD4 T cells infected in vitro with the HIV-1 virus strain 89.6 ($n = 3$ HIV- donors). Following in vitro infection, the expression of this transcript was upregulated at all time points tested compared to non-infected samples (median fold induction = 6.4, when considering all time points together) (Fig. 8A), and strongly correlated with the percentage of p24+ cells in each infected sample ($p < 0.0001$) (Fig. 8B). Furthermore, we sorted p24− and p24+ cells from three viremic individuals and measured *LINC02964* expression in these two populations: compared to p24− cells, the expression of *LINC02964* was on average 141 times higher in p24+ cells, confirming that active HIV infection is associated with the upregulation of *LINC02964* expression (Fig. 8C).

To assess the impact of *LINC02964* knockdown on active infection, CD4 T cells from 3 HIV- donors were incubated with five antisense oligonucleotides (ASOs) prior to in vitro infection with the HIV-1 virus strain 89.6 (Fig. 8D, E). For 3 out of 5 ASOs (ASO 1/2/3), *LINC02964* expression was downregulated by at least 50% compared to the non-targeting control ASO (NTC) (Fig. 8D). In parallel with the downregulation of *LINC02964* expression with ASOs 1–3, the percentage of p24+ cells was decreased compared to the NTC (means = 6.1% p24+ cells for NTC, and 2.3%, 1.6% and 2.8% p24+ cells for ASOs 1, 2 and 3, respectively), suggesting that *LINC02964* favors active HIV infection (Fig. 8E).

Finally, to study the role of *LINC02964* in latency reversal, we developed a stable SupT1 cell line carrying a latent single-round NL4.3-ΔENV-IRES-HSA. Although the incubation of the cells with ASOs 1/2/3 prior to Tat-LNP stimulation led to the downregulation of *LINC02964* expression (Fig. 8F), ASO treatment did not have an impact on latency reversal compared to the mock condition (means = 18.8% HSA+ cells in the mock condition, and 18.8%, 19.3%, 19.7% HSA+ cells for ASOs 1–3) (Fig. 8G). This observation indicates that, although *LINC02964* appears to be involved in active HIV infection, it does not appear to have a role in HIV reactivation.

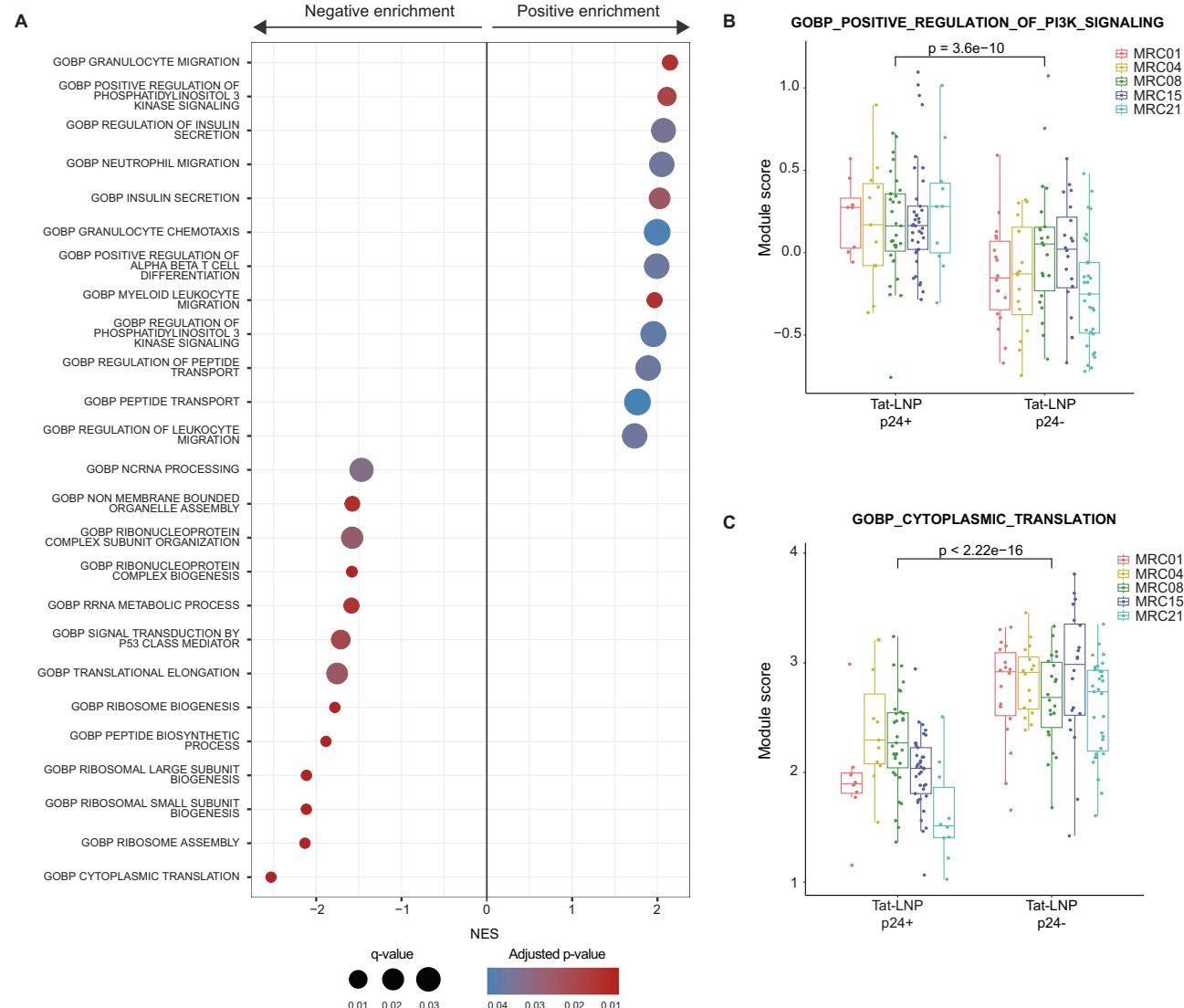

**Fig. 6 | Gene set enrichment analysis. A–C** CD4 T cells from *n* = 5 ART-treated individuals were stimulated for 48 h with Tat-LNP. p24− and p24+ cells were single-cell sorted and subjected to Smart-seq2. A ranked list of log2(fold change) was used as input for the "GSEA" function of the clusterProfiler package. **A** Dotplot showing the 25 most differentially regulated gene sets between p24+ and p24− cells (ranked by adjusted *p* value). NES normalized enrichment score (positive enrichment: upregulated in p24+ cells; negative enrichment: downregulated in p24+ cells). The list of significantly up/downregulated gene sets between p24+ and p24− cells is

shown is Supplementary Data 6. **B**, **C** Box plots showing the average expression levels for the "positive regulation of PI3K signaling" gene set (**B**), and for the "cytoplasmic translation" gene set (**C**), for each single cell from the p24− and p24+ fractions. Median values, interquartile ranges, minima and maxima are depicted on the graphs. For statistical analyses, two-sided non-parametric Mann–Whitney tests were used. The Gene Set Enrichment Analysis (GSEA) on the order ranked gene list can be found in Supplementary Data 6.

## Discussion

To our knowledge, we are the first to report a combination of LRAs that induces latency reversal in a higher fraction of latently infected cells than classic mitogens. The high reactivation capacity of this Tat-LNP/ PNB might reside in the distinct mechanisms of action of these two molecules: while PNB favors early events of HIV transcription by opening the chromatin, Tat promotes the elongation step by recruiting the P-TEFb complex to the LTR[32]. Measurement of p24 concentrations in the culture supernatants by SIMOA confirmed that Tat-LNP/PNB also enables the efficient release of viral particles in the supernatant, which is of particular interest in a clinical setting. When comparing the memory phenotype of p24+ cells between PMA/i and Tat-LNP/PNB-stimulated cells, there was a trend toward more p24+ cells with a TCM/TTM phenotype in the Tat-LNP/PNB condition, suggesting that combining LRAs with an HDACi could favor the reactivation of latent proviruses residing in CD4 T cells with a TCM phenotype,

as previously described[52]. By using the STIP-Seq assay[40], we showed that: (i) the proportions of intact and PSI/MSD-defective proviruses were similar between Tat-LNP/PNB and PMA/i-reactivated p24+ cells, with approximately 90% of the proviruses being PSI/MSD-defective and 10% being intact, (ii) the major infected clones observed following PMA/i stimulation were also retrieved in the Tat-LNP/PNB condition, with some rare and minor clones being retrieved in only one of the two conditions. In agreement with previous studies[53,54], we showed that MSD-defective proviruses can use alternative splice sites to produce spliced transcripts. The efficient production of Nef despite MSD defects would explain the observed downregulation of CD4 by p24+ cells carrying an MSD-defective provirus. Nevertheless, recent evidence suggests that PSI/MSD-defective proviruses exhibit limited infectivity due to reduced production of Env[54], indicating that they are unlikely to contribute to viral rebound after ART interruption. However, these proviruses can be the source of non-suppressible viremia in

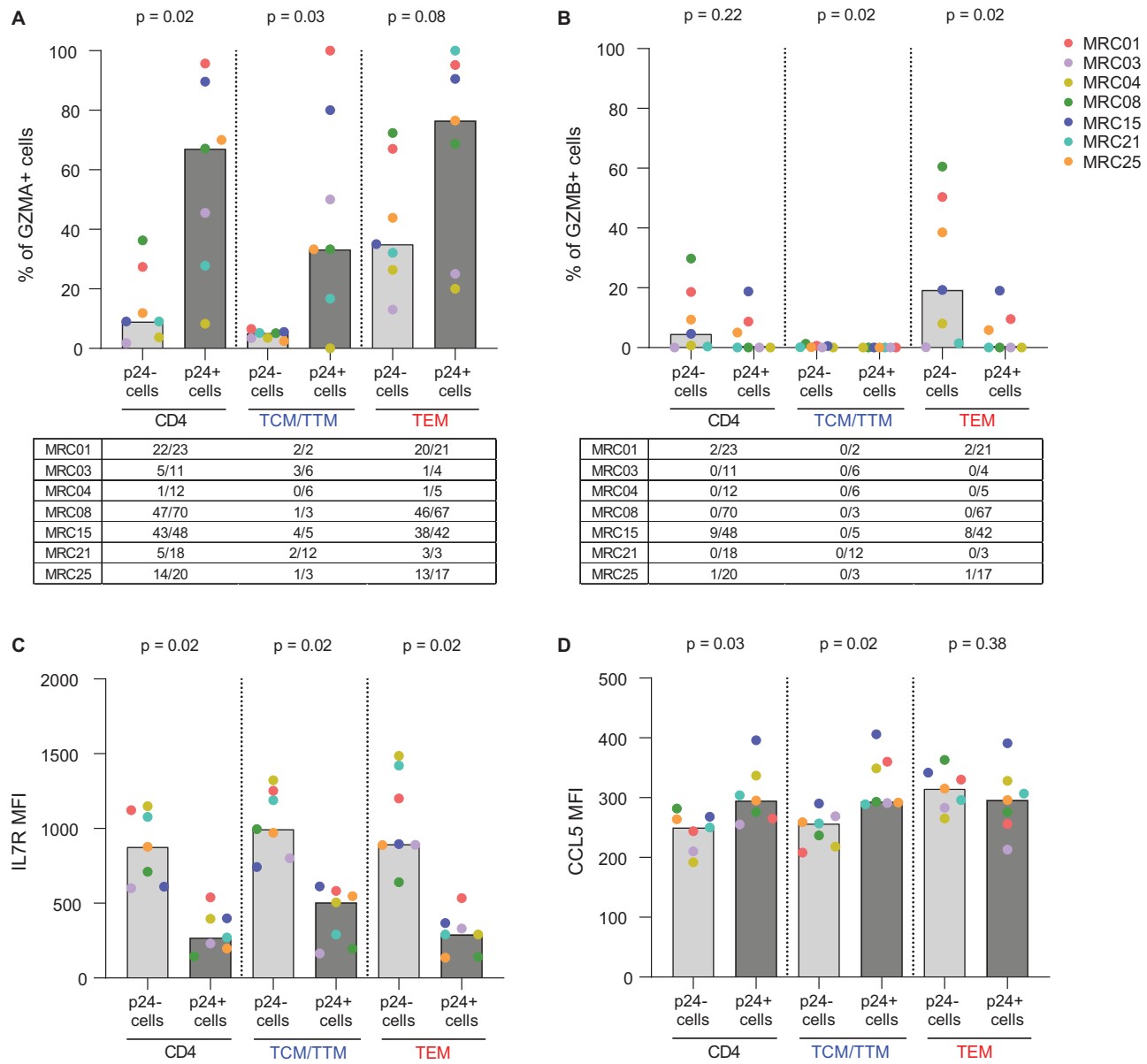

**Fig. 7 | Validation of the transcriptomic hits. A**−**D** CD4 T cells from *n* = 7 ART-treated individuals were stimulated for 48 h with Tat-LNP: GZMA, GZMB, IL7R, and CCL5 expression were assessed. **A**, **B** The percentage of GZMA+ cells (**A**) and GZMB+ cells (**B**) is compared between p24− cells versus p24+ cells. The tables below graphs show the number of p24+ cells that are GZMA or GZMB+ out of the total number of recorded p24+ cells for each analyzed fraction (all CD4 T cells, TCM/TTM, TEM). **C**, **D** The mean fluorescence intensity for IL7R (**C**) and CCL5 (**D**) is compared between p24− cells versus p24+ cells. Gray bars depict median values. For statistical analyses, two-sided non-parametric Wilcoxon tests were used. TN

naive T cells, TCM/TTM central and transitional memory T cells, TEM effector memory T cells, TTd terminally differentiated T cells. When a clear distinction between positive and negative subsets could be defined (GZMA, GZMB), results are expressed as a percentage of GZMA+/GZMB+ cells in the p24−/p24+ fractions; when a continuum of expression was observed with no clear distinction between positive and negative subsets (IL7R, CCL5), the results are expressed as IL7R/CCL5 MFI in the p24−/p24+ fractions (gating strategy in Supplementary Fig. 7A). Source data are provided with this paper.

ART-treated participants[54] and participate in chronic immune activation through the production of viral proteins[40,53,55], further justifying the need for a detailed characterization of reservoir cells carrying proviruses with PSI/MSD defects.

Furthermore, we demonstrated that Tat-LNP does not modify the transcriptome of CD4 T cells. Taking advantage of this unique property, we compared the transcriptomic features of p24+ cells to their p24− counterparts following Tat-LNP stimulation. While previous scRNA-seq studies compared the transcriptome of translation-competent reservoir cells to bulk CD4 non-infected cells[7,9], we took into account that most of the p24+ cells display a CD45RO+ memory

phenotype and therefore compared the transcriptome of p24+ cells to their memory p24− counterparts. This, along with the use of a compound that preserves the transcriptome of the CD4 T cells, might explain the more limited number but perhaps more relevant DEG we obtained in our analysis compared to previously described studies[7,9]. Among the most differentially expressed genes, we found that *LINC02964, SOD1P3, CCL5*, and *GZMA* were upregulated in p24+ cells, while *ATG10* and *IL7R* were downregulated in p24+ cells compared to p24− cells. To contextualize our findings, Supplementary Table 3 offers a comprehensive review of the latest scRNA-seq studies investigating the phenotypes of HIV-infected cells. As indicated in this table,

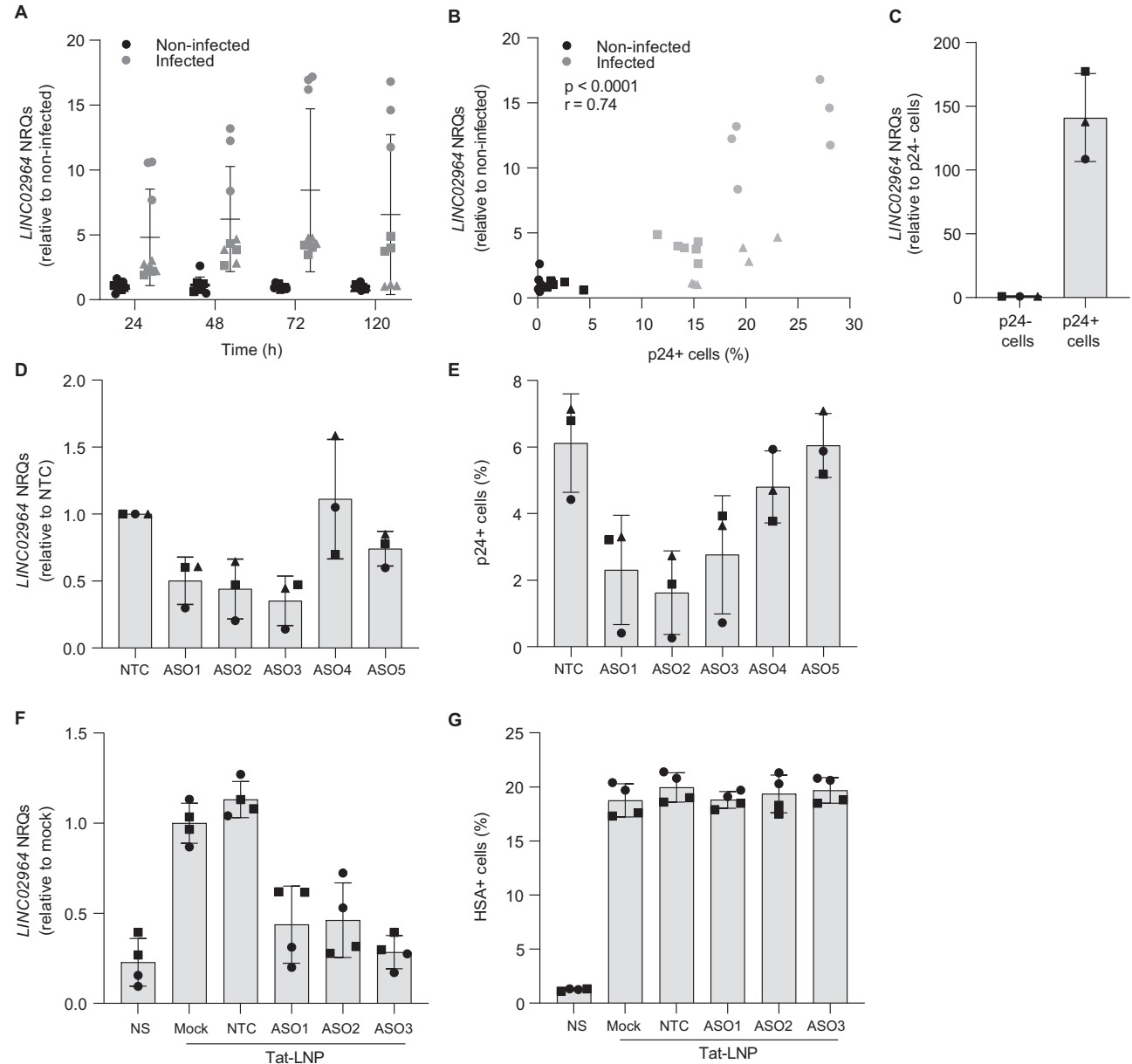

**Fig. 8 | Assessment of the role of *LINCO2964* in HIV-1 pathogenesis and latency reversal. A, B** Infection time course in CD4 T cells using the HIV strain 89.6 (*n* = 3 HIV- donors depicted by symbols, three technical replicates per condition). **A** Normalized relative quantities (NRQs) of *LINCO2964* obtained by RT-qPCR (expressed as fold inductions relative to non-infected samples). **B** Correlation plot between the expression levels of *LINCO2964* and the percentage of p24+ cells (defined by p24 KC57-RD1 intracellular staining) found at 48 h and 120 h post-infection. Non-parametric Spearman rank correlation test was performed (*p* = 0.0000003). **C** Bar plot showing *LINCO2964* NRQs in bulk sorted p24- and p24+ cells from 3 viremic donors, expressed as fold inductions relative to the "p24- cells" condition. **D, E** ASOs treatment of CD4 T cells prior to infection with the HIV

strain 89.6 (*n* = 3 HIV- donors depicted by symbols). Bar plot showing *LINCO2964* NRQs, expressed as fold inductions relative to NTC (**D**), and the percentage of p24+ cells defined by p24 KC57-RD1 intracellular staining (**E**), when using a non-targeting control ASO (NTC) and five ASOs targeting the last intron of *LINCO2964*. **F, G** ASOs treatment of SupT1 cells latently infected with the NL4.3-ΔENV-IRES-HSA lab strain prior to latency reversal with Tat-LNP (*n* = 2 independent experiments depicted by symbols, 2 technical replicates per condition). NS non-stimulated, Mock no ASO, NTC non-targeting control ASO. Bar plot showing *LINCO2964* NRQs, expressed as fold inductions relative to mock (**F**), and the percentage of HSA+ cells (**G**). Means and standard deviations are depicted on all graphs. Source data are provided with this paper.

and in agreement with our observation, the expression of cytotoxic T cell genes such as *GZMA* and *CCL5* was previously shown to be upregulated in HIV-1 RNA+ T cell clones compared to HIV-1 RNA- T cell clones in HIV viremic individuals[50], as well as in clones harboring intact HIV proviruses in ART-suppressed individuals[51]. Cytotoxic CD4 T cells detected in healthy human blood express high levels of the anti-apoptotic molecule Bcl-2[56], suggesting that cytotoxic CD4 T cells might have unique properties that favor their long-term persistence. Consistently, we found differential expression of *IL7R* and *ATG10*

between p24+ and p24- cells, two molecules involved in cell survival pathways. Gene set enrichment analysis further confirmed those observations, showing an upregulation of the PI3K signalling pathway in p24+ cells, a pathway directly involved in cell survival, metabolism and proliferation. Furthermore, we showed that gene sets associated with protein translation were downregulated in p24+ cells, which may subsequently protect HIV-infected cells against immune clearance by limiting virion production. These results are in agreement with a recent study showing HIV silencing and cell survival signatures in HIV-infected

memory CD4 T cells[57]. Further investigations are needed to determine whether these signatures are caused by HIV reactivation or if they reflect unique cellular traits that confer long-term survival benefits.

Finally, we confirmed the transcriptomic differences observed by Smart-seq2 at the protein level. In contrast with a recent publication by Collora et al.[50], which compared the phenotype of p24+ and p24− cells following PMA/i stimulation, we did not observe an enrichment of GZMB expression in the p24+ fraction compared to p24− cells. The use of different LRAs to reactivate HIV, PMA/i versus Tat-LNP, might explain this difference and highlights the importance of using LRAs that do not modify the phenotype of the cells. Finally, we confirmed the functional relevance of *LINCO2964* by showing the upregulation of this transcript in response to in vitro HIV infection of primary CD4 T cells. Additionally, the expression of this transcript was around 140 times higher in sorted p24+ cells compared to p24− cells in viremic individuals, further supporting the data obtained in vitro. The successful knockdown of *LINCO2964* expression using antisense oligonucleotides was associated with a reduction in the percentage of p24+ cells in in vitro infected CD4 T cells, suggesting that *LINCO2964* favours active HIV infection. In contrast, *LINCO2964* does not seem to play a role in latency reversal. Further investigation is required to pinpoint the exact role of this lncRNA in HIV pathogenesis and to elucidate its molecular mode of action.

In conclusion, Tat-LNP individually or in combination with other LRAs that preserve the phenotype of the cells, represents a valuable research tool for in vitro reservoir studies as it enables the characterization of the transcriptional landscape and the proteome of a high number of HIV reservoir cells in their near-native state, potentially leading to the discovery of new druggable targets. Although we restrained our FACS analyses to a limited number of proteins, future studies should involve mass cytometry or multi-color FACS, which allow the analysis of more than 30 markers simultaneously. Finally, future pre-clinical studies in humanized mice and/or non-human primate models are needed to assess biosafety in vivo and to confirm the reactivation capacity and subsequent reduction in reservoir size following Tat-LNP treatment.

## Methods

### Participants and blood collection
A total of $n = 23$ HIV-1 seropositive individuals on stably suppressive ART were included in this study (Supplementary Table 1). Participants were recruited at Ghent University Hospital. 2/23 individuals are female, 21/23 are male; the limited representation of female individuals in our study is a direct reflection of the population of HIV-1 seropositive individuals in Belgium, which predominantly consists of men who have sex with men. Participants underwent leukapheresis to collect large numbers of peripheral blood mononuclear cells (PBMC). PBMCs were isolated by Ficoll density gradient centrifugation and were cryopreserved in liquid nitrogen.

### Inclusion and ethics statement
All participants were adults and provided written informed consent. All three studies and their study protocols were approved by the Ethics Committee of the Ghent University Hospital (Belgium) (EC numbers: STAR 2015/0894[58]; ISALA 2015/0771[59]; Mercuri BC-07056). All MRC participants were recruited as part of the present study (Mercuri, NCT04305665). STAR10 and UZG3034 were recruited as part of the STAR (NCT02641756) and ISALA (NCT02590354) studies, respectively. These two participants exhibited elevated frequencies of p24+ cells upon stimulation. Their inclusion further expanded the pool of participants for our investigations (e.g. phenotype of p24+ cells and STIP-Seq/Smart-seq2 analyses). Analyses presented in this manuscript are covered by the three protocols. There was no experimental drug administration performed in any of these studies.

### Antibodies
Fixable Viability Stain 510 was obtained from ThermoFisher Scientific (#L34957, 1/1000). The following antibodies were used in staining experiments: CD3 AF700 Clone UCHT1 (BD Biosciences, #557943, 1/50), CD4 PE-Cy7 Clone L200 (BD Bioscience, #560644, 1/200) and CD4 BV786 Clone SK3 (BD Biosciences, #563881, 1/200), CD8 BV510 Clone RPA-T8 (BioLegend, #301047, 1/200), CD45RO BV421 Clone UCHL1 (BD Biosciences, #562649, 1/100) or CD45RO PE Clone HI100 (BD Biosciences, #5555493, 1/20), CD27 BV605 Clone L128 (BD Biosciences, #562656, 1/100), PD1 BB700 Clone EH12.1 (BD Biosciences, #566461, 1/100), CD69 PerCP-Cy5.5 Clone FN50 (BD Biosciences, #560738) and CD69 APC (BioLegend, #310909, 1/50), CD25 BV421 Clone M-A251 (BD Biosciences, #562443, 1/50), HLA-DR FITC Clone REA805 (Miltenyi Biotech, #130-111-788, 1/50), GZMA PE-Cy7 Clone CB9 (BioLegend, #507221, 1/25), GZMB PB Clone GB11 (BioLegend, #515407, 1/50), CCL5 PerCP-Cy5.5 Clone VL1 (BioLegend, #515507, 1/50), CD127 PE-CF594 Clone HIL-7R-M21 (BD Biosciences, #562397, 1/50). For p24 staining, we used a combination of two antibodies: p24 KC57-FITC (Beckman Coulter, #6604665, 1/500) and p24 28B7-APC (MediMabs, #MM-0289-APC, 1/400).

### Negative selection of CD4 T cells
CD4 T cells were isolated from PBMC by negative magnetic selection using the EasySep Human CD4 T Cell Enrichment Kit (StemCell Technology, #19052). Purity was typically >98%.

### CD4 T cell stimulation
CD4 T cells were resuspended at $2 \times 10^6$ cells/mL in RPMI + 10% Fetal Bovine Serum and antiretroviral drugs were added to the culture (200 nM raltegravir, 200 nM lamivudine) to avoid new cycles of replication. Cells were rested for at least an hour at 37 °C before being stimulated with the following LRAs: 1 µg/mL ionomycin (Sigma, #I9657) and 162 nM PMA (Sigma, #P8139), 50 nM panobinostat (Selleckchem, LBH589), Tat-LNP (250 ng/mL; 1.4 nM). When combined, the same concentrations were used for Tat-LNP and PNB.

### Microarray processing
Following a 6 and 24 h-stimulation of bulk CD4 T cells with the LRAs of interest and in presence of antiretroviral drugs, 10,000 cells were transferred to a tube containing 100 µL of Buffer RLT (Qiagen, #1015762) supplemented with 1% 2-mercaptoethanol (Sigma-Aldrich, #M3148). Cells were stored at −80 °C until further processing. RNA extraction was done using the RNeasy plus mini kit (Qiagen, #74181). Amplification and biotin labeling of total RNA were performed using the GeneChip PICO Reagent Kit following the manufacturer's instructions (ThermoFisher, #902790). Biotin-labeled target samples were hybridized to the Clariom GO Screen (ThermoFisher, #952361) containing probes for over 20,000 genes. Target hybridization was performed on the GeneTitan Multi-Channel Instrument (ThermoFisher, #00-0373) according to manufacturer's instructions provided for Expression Array Plates. Images were analyzed using the GeneChip Command Console (GCC) Software (v6.1.1.195; ThermoFisher).

### Microarray data analysis
The microarray datasets were background corrected and normalized by robust multiarray analysis and summarized with the ClariomSHumanHT_Hs_ENTREZG v21.0.0 chip definition files. Quality assessment was performed with arrayQualityMetrics. Using the expression values for all 22,593 probe sets, we evaluated the main source of variability using an Euclidean distance matrix. In 2-dimensional space, the Euclidean distance between two points (Point 1 and Point 2) is calculated from the cartesian coordinates of the points (x1, y1) and (x2, y2). The Euclidean distance is obtained by: $\sqrt{(x1-x2)^2 + (y1-y2)^2}$. We identified three distinct clusters based on stimulation: cluster 1

(NS samples or treated with DMSO, HA-LNP, Tat-LNP); cluster 2 (Tat-LNP/PNB and PNB) and cluster 3 (PMA/i). We performed DGE analysis (i.e. clusters 2 and 3 were compared to cluster 1) using linear regression models for microarray data analysis (Limma) (Supplementary Data 3 and 4). Genes identified were considered statistically significant based on Benjamini−Hochberg adjusted $p$ value < 0.01. We carried out GSEA to assess gene set enrichment on the order ranked gene list (ranked based on log2 fold changes; DGE analysis results output) using ClusterProfiler (v3.16.1) (Supplementary Data 3 and 4). Pathways and terms identified were deemed statistically significant based on Benjamini−Hochberg adjusted $p$ value < 0.05.

**P24-SIMOA**

2−5 million CD4 T cells were stimulated for 24 h and 48 h with the LRAs of interest in presence of antiretroviral drugs (200 nM raltegravir, 200 nM lamivudine). Following stimulation, supernatants were collected and stored at −80 °C until further processing. Supernatants were thawed at room temperature and concentrations of Gag p24 were determined on a SIMOA HD-1 analyzer using a validated SIMOA p24 kit (Quanterix, USA) following the manufacturer's instructions. p24 concentrations were calculated from raw signal average enzyme per bead (AEB) using a four-parameter logistic regression curve fitting.

**HIV-Flow procedure**

Frequencies of p24-producing cells following LRA stimulation were measured by using a combination of 2 antibodies targeting the p24 protein (p24 KC57-FITC, p24 28B7-APC) as previously described by Pardons et al.[8]. The detailed protocol of the HIV-Flow procedure can be found here: https://doi.org/10.17504/protocols.io.w4efgte.

**Methanol-based HIV-Flow procedure for STIP-Seq**

The methanol-based HIV-Flow procedure was performed as previously described by Cole et al.[40]. The detailed protocol can be found here: https://protocols.io/view/methanol-based-hiv-flow-bpedmja6. In brief, following stimulation, a maximum of $5 \times 10^6$ cells per condition were resuspended in PBS and stained with fixable viability stain 510 for 20 min at RT. Cells were then stained with antibodies against cell surface molecules (CD3, CD4, CD8, CD45RO, CD27, PD1) in PBS + 2% FBS for 20 min at 4 °C. After a 5 min-centrifugation step at 4 °C to pre-chill the cells, CD4 cells were vortexed to avoid clumping and 900 μL of ice-cold methanol (−20 °C) was gently added. Cells were fixed/permeabilized in methanol for 15 min on ice. Intracellular p24 staining was performed using a combination of 2 antibodies (p24 KC57-FITC, p24 28B7-APC) (45 min, RT) in PBS/1% BSA (ThermoFisher, #AM2616)/RNAse inhibitor 0.4 U/μL (Promega, #N2615)/DTT 1 mM. All washing steps following methanol permeabilization were done in PBS/0.04% BSA. In all experiments, CD4 T cells from an HIV-negative control were included to set the threshold of positivity.

**Methanol-based HIV-Flow procedure for single-cell RNA-seq**

The HIV-Flow procedure used for single-cell RNA-seq experiments was similar to the one described below, with some exceptions. All surfaces and materials were cleaned with RNAse away (ThermoFisher, #7003) to prevent from RNA degradation. Following fixation/permeabilization with methanol, all washing steps were done in PBS/0.04% BSA (ThermoFisher, #AM2616)/RNAse inhibitor 0.2 U/μL (Promega, #N2615)/DTT 1 mM. Cells were resuspended in the same buffer before sorting. All tubes were kept on ice and protected from light until acquisition and sorting.

**Single cell sorting of p24+ cells**

Single p24+ cells were sorted on a BD FACSAria Fusion Cell Sorter. Cells were sorted in skirted 96-well PCR plates (Biorad, #12001925). To avoid evaporation during the sort, the PCR plate was continuously chilled at 4 °C. For STIP-Seq experiments, cells were sorted into a volume of 4 μL PBS sc 1X (Qiagen, #150345). For single-cell RNA-seq experiments, cells were sorted into 2 μL lysis buffer, consisting of 0.1 μL RNAse inhibitor (Takara, #2313B) and 1.9 μL 0.2% vol/vol Triton X-100 (Sigma-Aldrich, #T8787). Besides, for each participant, 20 p24− TCM/TTM cells and 20 p24− TEM cells were sorted. Index sorting, a procedure where coordinates of single-sorted cells for all markers are documented, was used to enable phenotyping of single-sorted p24+ cells. CD4 T cell memory subsets were defined as follows: TN = CD45RO− CD27+, TCM/TTM = CD45RO + CD27 +, TEM = CD45RO + CD27−, TTd = CD45RO− CD27−. Flow-Jo software v10.6.2 was used to analyze flow cytometry data (Tree-Star).

**Multiple displacement amplification (MDA)**

Whole genome amplification of single-sorted cells was carried out by multiple displacement amplification with the REPLI-g single-cell kit (Qiagen, #150345), according to manufacturer's instructions. A positive control, consisting of 10 p24− cells sorted into the same well, was included on every plate.

**Quantitative polymerase chain reaction (qPCR) for RPP30**

After whole genome amplification by MDA, reactions were screened by qPCR on the RPP30 reference gene, as described previously[40]. Reactions that yielded a cycle of threshold (Ct) value of 38 or lower, were selected for further downstream processing.

**Integration site analysis**

MDA reactions that were positive for RPP30 were subjected to integration site sequencing by a modified version of the integration site loop amplification (ISLA) assay, as described previously[40]. Resulting amplicons were visualized on a 1% agarose gel and positives were sequenced by Sanger sequencing. Analysis of the sequences was performed using the 'Integration Sites' webtool (https://indra.mullins.microbiol.washington.edu/integrationsites).

**Near full-length proviral genome amplification**

MDA wells that were RPP30 positive were subjected to 5' and 3' half genome amplification. The 25 μL PCR mix for the first-round is composed of: 5 μL 5× Prime STAR GXL buffer, 0.5 μL PrimeStar GXL polymerase (Takara Bio, #R050B), 0.125 μL ThermaStop (Sigma Aldrich, #TSTOP-500), 250 nM forward and reverse primers, and 1 μL of 1/5 diluted MDA product. The mix for the second round has the same composition and takes 1 μL of the first-round product as an input. Thermocycling conditions for first and second PCR rounds are as follows: 2 min at 98 °C; 35 cycles (10 sec at 98 °C, 15 sec at 62 °C, 5 min at 68 °C); 7 min at 68 °C. For MDA wells that did not yield a 5' or 3' amplicon, 2 pairs of primers (Fragment 1 and Fragment 2 for the 5' amplicon[60]; A2 and B2 for the 3' amplicon[61]) were used on the 1st round product. The primer sequences for the different approaches are summarized in Supplementary Table 4. The summary of all PCRs that were performed is shown in Supplementary Data 1.

**Near full-length proviral genome sequencing and assembly**

Amplicons were pooled at equimolar ratios and cleaned by magnetic bead purification (Ampure XP, Beckman Coulter, #A63881), followed by quantification with the Quant-iT PicoGreen dsDNA Assay Kit (Invitrogen, #P7589). Library preparation was done with the Nextera XT DNA Library Preparation Kit (Illumina, #FC-131-1096) according to manufacturer's instructions, with indexing of 96-samples per run. The library was sequenced on a MiSeq Illumina platform via 2 × 150 nt paired-end sequencing with the 300 cycle v2 kit (Illumina, #MS-102-2002) according to manufacturer's instructions, yielding ~200,000 reads per sample. Near full-length proviral genome sequences were de novo assembled as follows: (1) FASTQ quality checks were performed

with FastQC (v0.11.7, http://www.bioinformatics.babraham.ac.uk/projects/fastqc) and removal of Illumina adaptor sequences and quality-trimming of 5′ and 3′ terminal ends was performed with BBtools (v37.99, sourceforge.net/projects/bbmap/). (2) Trimmed reads were de novo assembled using MEGAHIT (v1.2.9)[62] with standard settings. (3) Resulting contigs were aligned against the HXB2 HIV-1 reference genome using blastn (v2.7.1)[63] with standard settings, and contigs that matched HXB2 were retained. (4) Trimmed reads were mapped against the de novo assembled HIV-1 contigs to generate final consensus sequences based on per-base majority consensus calling, using bbmap (v37.99, sourceforge.net/projects/bbmap/). Scripts concerning de novo assembly of HIV-1 genomes can be found at the following GitHub page: https://github.com/laulambr/virus_assembly[64].

## Proviral genome classification

NFL proviral genome classification was performed using the Proviral Sequence Annotation & Intactness Test (ProSeq-IT) (https://psd.cancer.gov/tools/tool_index.php) and the publicly available 'Gene Cutter' and 'Hypermut' webtools from the Los Alamos National Laboratory HIV sequence database (https://www.hiv.lanl.gov). Proviral genomes were classified in the following sequential order: (1) 'Inversion': presence of internal sequence inversion, defined as region of reverse complementarity. (2) 'Large internal deletion': internal sequence deletion of >1000 bp. (3) 'Hypermutated': APOBEC-3G/3F-induced hypermutation. (4) 'PSI/MSD defect': deletion >7 bp covering (part of) the packaging signal region, or absence of GT dinucleotide at the MSD and GT dinucleotide at the cryptic donor site (located 4 bp downstream of MSD)[65]. Proviruses with a deletion covering PSI/MSD that extended into the gag gene - thereby removing the gag AUG start codon - were also classified into this category. (5) 'Premature stop-codon/frameshift': premature stop-codon or frameshift caused by mutation and/or sequence insertion/deletion in the essential genes gag, pol or env. Proviruses with insertion/deletion >49 nt in gag, insertion/deletion >49 nt in pol, or insertion/deletion >99 nt in env were also classified into this category. (6) 'Intact': proviruses that displayed none of the above defects were classified into this category.

## Single-cell RNA-seq by Smart-seq2

Following sorting of single p24+ and p24− cells into 2 µL of lysis buffer, 96-well plates were flash-frozen on dry ice and stored at −80 °C until further processing. Plates were thawed and amplified cDNA was generated through Smart-seq2, as described by Picelli et al.[47,48], with one modification. To minimize the formation of primer dimers and/or other concatemers, the concentration of the oligo-dT was lowered tenfold, from 10 µM to 1 µM. This modification removed all concatemers, while retaining library complexity. Following cDNA generation, 24 cycles of pre-amplification were performed to generate sufficient yield for library preparation. Libraries were generated for each individual cell using the Nextera XT DNA Library Preparation Kit (Illumina, #FC-131-1096) according to manufacturer's instructions, with indexing of 384 cells per run. Libraries of 384 cells were pooled equimolarly and sequenced on a Nextseq 500 via 2 × 150 nt paired-end sequencing with the NextSeq 500/550 High Output Kit v2.5 (300 Cycles) (Illumina, #20024908), yielding ~ 1,000,000 reads per cell. Smart-seq2 processing yielded a total of 108 (20 TCM/TTM, 88 TEM) and 212 (6 TN, 124 TCM/TTM, 79 TEM, 3 TTd) p24+ cells following treatment with Tat-LNP alone or combined with PNB, respectively. The median number of p24+ cells per participant was 14 for Tat-LNP and 30 for Tat-LNP/PNB (Supplementary Data 2). In parallel, a total of 109 (59 TCM/TTM, 50 TEM) and 150 (95 TCM/TTM, 55 TEM) p24− cells were sorted after stimulation with Tat-LNP or Tat-LNP/PNB, respectively. The median number of p24− cells per participant was 20 for Tat-LNP and 21 for Tat-LNP/PNB (Supplementary Data 2). In addition, a total of 28 PMA/i-stimulated p24+ cells and 17 NS cells were sorted as positive and negative controls, respectively.

## Single-cell RNA-seq analysis

The overall quality of fastq files was assessed by FastQC (http://www.bioinformatics.babraham.ac.uk/projects/fastqc) and removal of Illumina adaptor sequences and quality-trimming of 5′ and 3′ terminal ends was performed with BBtools (v37.99, sourceforge.net/projects/bbmap/). Further information on the QC metrics is provided in Supplementary Table 5. Trimmed reads were aligned to the human reference genome GRCh38/hg38 using the STAR aligner (v2.7.10a)[66]. Per-gene read counts were obtained with the Featurecounts function of the Subread package (v2.0.3)[67]. Filtering of low-quality cells was performed by removing all cells with less than 200 or more than 4500 detected genes, as well as cells with >20% reads mapping to mitochondrial genes. Raw read counts were normalized for read depth by ln(CPM + 1) transformation, and UMAP dimensionality reduction was performed using the Seurat (v4.1.0) package[68] in R (v4.2.0). Normalized read counts were used to infer differentially expressed genes (DEG) using the MAST (v1.20.0) algorithm[69] in R (v4.2.0). p-values were derived from likelihood ratio tests. p-values lower than 0.05 after Bonferroni corrections were considered significant.

## Pathway enrichment analysis

GSEA was performed with the "clusterProfiler" (v4.8.0) package in R. Gene Ontology gene sets were downloaded from MsigD[70], while canonical pathway gene sets combine gene sets from the Wiki-Pathways, Reactome, KEGG, PID and BioCarta databases. Gene sets with a size <20 or >500 genes were excluded from the analysis. A ranked list of log2(fold change) was used as input of the "GSEA" function of the clusterProfiler package. The 25 most significant pathways (ranked by adjusted p value) were subsequently visualized with the "dotplot" function of the "enrichplot" (v1.20.0) R package. The AddModuleScore function in the Seurat R package was used to assign a score for each gene set of interest to each single cell. After binning all genes into n = 20 bins based on average expression, n = 100 control genes from the same bin were randomly sampled for each gene in the gene set. The average expression of the control genes was subtracted from the average expression levels of each gene set for each single cell.

## Viral genome assembly from single-cell RNA-seq data

For de novo assembly, the RNA-seq reads from each HIV-1 infected cell were trimmed and mapped to the human reference genome GRCh38/hg38 using STAR aligner (v2.7.10a)[66] to remove host reads. These host-depleted reads were de novo assembled using MEGAHIT (v1.2.9)[62] with standard settings. The resulting contigs were aligned against the HXB2 HIV-1 reference genome using blastn (v2.7.1)[63] with standard settings, and contigs that matched HXB2 were retained. Trimmed host-depleted reads were mapped against the de novo assembled HIV-1 contigs to generate final consensus sequences based on per-base majority consensus calling, using bbmap (v37.99, sourceforge.net/projects/bbmap/).

For each participant, STIP-Seq proviruses (if available) and RNA-seq viruses were aligned to HXB2 using MAFFT (v7.471)[71]. These alignments were manually inspected and categorized as clones in the following sequential order: 1) RNA-seq viruses being 100% identical to STIP-Seq proviruses, 2) If incomplete coverage of the RNA-seq viral sequence, RNA-seq viruses spanning the same identical deletion junction sites as seen in STIP-Seq proviruses, and 3) RNA-seq viruses not matching STIP-Seq proviruses, but being 100% identical or spanning the same identical deletion junction sites in the RNA-seq data are labeled as expansions of identical sequences (EIS).

To construct the phylogenetic tree, fully covered STIP-Seq and RNA-seq sequences were aligned using MAFFT (v7.471)[71]. The phylogenetic tree was constructed using PhyML v3.0[72] (best of NNI and SPR rearrangements) and 1000 bootstraps. For more clarity, sequences belonging to a clone are only included once per assay.

Scripts concerning de novo assembly of HIV-1 genomes from RNA-seq data will be uploaded on GitHub upon acceptance of this manuscript.

## HIV splice site analysis

Trimmed, host-depleted RNA-seq reads from HIV-1 infected cells were mapped using the STAR aligner (v2.7.10a) to their respective constructed RNA-seq viruses. If cells were identified as being part of an infected cell clone, the most complete de novo RNA-Seq virus was used as reference. Splice junctions were detected in each cell using STAR, after which they were inspected and manually annotated in Geneious (v2020.2). We focused our analysis on the detection of alternative donor sites to the major splice donor site (MSD, D1), as the MSD was mutated or deleted in the majority of proviral genomes recovered from p24+ cells. Figure 4C shows splice donor sites in the 5' UTR region that are used by HIV-1 proviruses to splice into known acceptor sites (A1-5).

## Assessment of *LINC02964* expression following in vitro infection

CD4 T cells from three HIV- donors were positively isolated using CD4 MicroBeads (Miltenyi Biotec, #130-045-101) and resuspended at 1x10^6 cells/mL in RPMI + 10% Fetal Bovine Serum + 20 ng/ml Human IL-2 (PeproTech, #200-02-250UG). CD4 T cells were stimulated with anti-CD3/CD28 beads (1:100 dilution, Miltenyi Biotec, #130-111-160) for 2 days before infection with the HIV-1 strain 89.6 (NIH HIV Reagent Program, Division of AIDS, NIAID, #ARP-3552) at a multiplicity of infection of 1. Infection was performed for each donor in triplicate. RNA extraction was performed using the InnuPREP RNA mini kit 2.0 (#845-KS-2040250) following manufacturer's instructions. RNA concentrations were measured using the Nanodrop™ 2000 (Thermo-Fischer Scientific) and 200 ng of total RNA was used in the reverse transcription reaction using qScript cDNA Supermix (Quantabio, #95048-500) following manufacturer's protocol. qPCR was performed in duplicates on 5–10 ng of the synthetized cDNA to assess the expression of *LINC02964* and four reference genes (*ACTB, GAPDH, β2M, YHWAZ*). Composition of the PCR mix was as follows: 250 nM forward and reverse primers, 2.5 μL LightCycler480 SYBR Green I Master (Roche Diagnostics, #04707516001) in a total reaction volume of 5 μL. Thermocycling conditions were as follows (LightCyler 480; Roche Applied Science): 5 min at 95 °C; 45 cycles (10 sec at 95 °C, 15 sec at 60 °C, 15 sec at 72 °C); melting curve from 60 to 95 °C and cooldown at 40 °C for 1 min. Normalized relative quantities (NRQs) were calculated using the qbasePLUS software (Biogazelle, Belgium), and were scaled to the corresponding uninfected control samples for each donor.

## Assessment of *LINC02964* expression in p24+ and p24− cells from viremic individuals

Following resting overnight, CD4 T cells from three viremic individuals were intracellularly stained for p24 (p24 KC57-FITC, p24 28B7-APC) following the methanol-based HIV-Flow procedure. A minimum of 100 p24+ cells and 700 p24− cells were bulk sorted in 96 wells plate containing 5 μL Single Shot lysis buffer (Biorad, #1725080). cDNA was made using the iScript cDNA synthesis kit (Biorad, #1708890), following the manufacturer's instructions. The expression of *LINC02964* was assessed by qPCR, as described above. NRQs were calculated using the qbasePLUS software (Biogazelle, Belgium), and were scaled to the corresponding "p24− cells" condition for each donor.

## Knockdown of *LINC02964* expression using antisense oligonucleotides (ASOs)

CD4 T cells from three HIV- donors were incubated for 2 days with five ASOs (5 μM) targeting the last intron of *LINC02964*, and with a non-targeting control ASO (NTC). CD4 T cells were then infected with the

HIV-1 strain 89.6 (NIH HIV Reagent Program, Division of AIDS, NIAID, #ARP-3552). Two days post-infection, we assessed the fraction of cells expressing p24 by flow cytometry (p24 KC57-PE) and validated the knockdown of *LINC02964* expression by RT-qPCR as described above. *LINC02964* expression levels (NRQs) were scaled to the NTC condition of each donor.

| ASO | Sequence |
|---|---|
| NTC | TCATACTATATGACAG |
| ASO 1 | TGCTCCAGACGTCTGT |
| ASO 2 | GCGAAGGGCTGGTACC |
| ASO 3 | ACCGGAAACAGGCTGA |
| ASO 4 | GCCATGCAATGGTGGT |
| ASO 5 | AACTGCTGCAGTAGGA |

## Assessment of *LINC02964* expression in latently infected SupT1 cells

SupT1 cells (NIH Reagent Program, #ARP-100) were infected with a single-round NL4.3-ΔENV-IRES-HSA lab strain. The surface expression of HSA was assessed by flow cytometry every three days: three weeks post-infection, the percentage of p24+ cells decreased from 44.7% to 2%. Latently infected SupT1 cells were incubated for 2 days with ASOs 1/2/3 (1 μM), and with a non-targeting control ASO (NTC). Cells were then stimulated or not with Tat-LNP (250 ng/mL; 1.4 nM). Following overnight stimulation, we assessed the fraction of cells expressing HSA by flow cytometry and validated the knockdown of *LINC02964* expression by RT-qPCR as described above. *LINC02964* expression levels (NRQs) were scaled to the mock condition. Two independent experiments were conducted, and two technical replicates were performed per condition.

## Data representations and statistical analyses

Bar and line charts were generated with Graphpad Prism (v8.0.2). Violin plots were generated in R (v4.1.1) using the ggplot2 package (v3.3.5) and the "geom_sina" function from the ggforce package (v0.3.3). The correlation matrix on Supplementary Fig. 5A was generated in R (v3.6.1) using the "dist" function from the stats package (v4.0.2), as well as the ComplexHeatmap package (v2.4.3). The gene ontology heatmap on Supplementary Fig. 5B was generated in R (v3.6.1) using the ComplexHeatmap package (v2.4.3). The heatmap on Supplementary Fig. 6B was generated in R (v4.2.0) using the ditto-Heatmap function of the dittoSeq package (v1.8.1). The phylogenetic tree on Fig. 4A was visualized in R (v4.1.1) using the ggtree (v3.2.1)[73], ape (v5.6)[74] and the phytools (v1.0-1)[75] packages.

The median fold induction between two paired groups was obtained by calculating the fold induction for each individual participant and reporting the median of those fold inductions (Fig. 1). For correlation testing, the Spearman's rank correlation coefficient was calculated (Fig. 8B, Supplementary Fig. 4B, C). For paired group comparisons, two-sided non-parametric Wilcoxon tests were used. For non-paired group comparisons, two-sided non-parametric Mann-Whitney tests were used. For pairwise comparisons between groups with unequal variances but normal distributions, the Games-Howell test was used followed by a Bonferroni correction for multiple comparisons. To generate Supplementary Fig. 4B, C, p24 antibody fluorescence intensities and the percentage of HIV reads were normalized using a z-score (standard score). *P* values lower or equal to 0.05 were considered statistically significant.

## Availability of materials

For all requests pertaining to the utilization of the Tat-LNP molecule, kindly direct your inquiries to Janssen's Infectious Diseases and

Diagnostics Department (Beerse, Belgium). Contact person: Ellen Van Gulck.

## Reporting summary

Further information on research design is available in the Nature Portfolio Reporting Summary linked to this article.

## Data availability

The HIV-1 near full-length sequences generated in this study have been deposited in the Genbank database under accession codes OQ596882-OQ596960. Smart-seq2 data generated in this study have been deposited on SRA under accession code PRJNA888069. Micro-array data generated in this study have been deposited on GEO under accession code GSE247402. Tables that cannot fit onto three 8.5" × 11" pages are provided as Supplementary data files (Supplementary Data 1–6). Source data are provided with this paper.

## Code availability

Scripts concerning de novo assembly of HIV-1 genomes are freely available on GitHub: https://github.com/laulambr/virus_assembly[64]. A description of the key operations and instructions on how to install and run the code are provided on the GitHub page. In addition, a test dataset and a description of expected outputs with expected run times are provided.

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

## Acknowledgements

We thank all participants who donated blood samples, as well as MDs and study nurses who helped with the recruitment and coordination of this study. We are particularly thankful for the precious help received from Jerome Wayet and Anne Van den Broeke (Liège University) to set up the Smart-seq2 assay in our laboratory. The study team thanks Sophie Vermaut and Maarten Verdonckt for assisting with the flow cytometry sorting experiments, as well as Bram Parton and Kim De Leeneer for helping with the Illumina sequencing (Gent, Belgium). We thank the sequencing and the microarray cores from Janssen (Beerse, Belgium). Finally, we thank Jinho Park from Arcturus Therapeutics for formulating the LNP. This current research work was supported by VLAIO O&O (HBC.2018.2278). M.P. and S.R. were supported by postdoctoral funding from VLAIO O&O (HBC.2018.2278). B.C. and L.L. were supported by FWO Vlaanderen (1S28918N, 1S29220N, respectively). W.V.S. was supported by FWO-SBO SAPHIR grant (2020000501) and FWO Junior grant (G0B3820N). LV was supported by the Research Foundation Flanders (1.8.020.09.N.00) and the Collen-Francqui Research Professor Mandate.

## Author contributions

M.P., B.C. and L.V. conceptualized the experiments. M.P., B.C., Y.N. and W.V.S. performed experiments. F.E. did all analyses and figures related to the microarray data. L.L. analyzed all viral sequencing data and constructed phylogenetic trees. S.R., N.D.L. and A.D. helped with the ethics committee's approval and collection of samples. E.V.G. and E.N. helped with the coordination of the study, scientific input, and p24-SIMOA experiments. D.B. created the Tat RNA molecule. J.V. contributed to the design and formulation of the lipid nanoparticle. M.P. and B.C. wrote the paper. All authors read and edited the paper.

## Competing interests

The authors M.P., B.C., L.L., W.V.S., S.R., Y.N., N.D.L., A.D. and L.V.D. declare no competing interests. The authors E.V.G., E.N., F.E. and D.B. declare the following competing interests: these authors are employees of Johnson & Johnson and may be Johnson & Johnson, stockholders. The author J.V. declares the following competing interests: this author was an employee of Arcturus Therapeutics and may be an Arcturus Therapeutics stockholder.
