## [Peer Review File · Nature Communications]

Reviewers' Comments:

Reviewer #1:

Remarks to the Author:

In the manuscript from Pardons et al., entitled " Potent Latency Reversal by TAT RNA Enables Multi-Omic Analysis of the HIV-1 Reservoir" explored the HIV-1 reactivation capacity of a lipid nanoparticle containing Tat mRNA (Tat-LNP) in CD4 T cells from ART-treated individuals. The Authors found that the Tat-LNP, when combined with panobinostat, was able to induce HIV-1 latency reversal at a higher proportion of latently infected cells compared to PMA/ionomycin(≈4-fold).

Gene expression analysis showed that Tat-LNP alone does not significantly alter the transcriptome of CD4 T cells. The Authors identified transcriptomic differences between infected cells carrying an inducible provirus and non-infected cells, including the upregulation including GZMA, CCL5 and a long non-coding RNA LINC02964. Moreover, p24+ cells exhibit heightened PI3K/Akt signaling.

General comments

The Authors are showing not very novel as data as validation of their tat Mediated HIV-activation strategy has been already shown by others and even by the same Authors but sent to another. Besides the lack of novelty of their main finding, the remaining data presented in this manuscript is merely observational with no real understanding of the biology behind. Paradigmatic are the results obtained by scRNA sequencing, which were confined to very few cells, thus reducing the confidence of this reviewer on the solidity of their results.

The activation of a long non-coding RNA LINC02964 finding is potentially interesting, but no data at all on the functions is provided.

Altogether, I consider a low priority paper for Nature Communications.

Reviewer #2:

Remarks to the Author:

Summary of findings:

Pardons et. al. examined the reactivation potential of Tat mRNA encapsulated in a lipid nanoparticle (Tat-LNP) and used it to study the properties of HIV-infected CD4 T cells from ART treated patients. In combination with an HDAC inhibitor Panobinostat (PNB), Tat-LNP was found to be more effective in reversing latency but less cytotoxic compared with PMA/ionomycin (PMA/i). After stimulation with Tat-LNP/PNB significant enrichment of p24+ cells were found in the effector memory population compared to the Naïve or terminally differentiated cells.

Integration site as well proviral sequences between different conditions from five participants on ART were identified. Overall, proviral landscapes were similar between Tat-LNP/PNB and PMA/i conditions. Higher degree of latency reversal was noted after treating with Tat-LNP/PNB than with Tat-LNP alone as shown by the HXB2 mapped reads.

The study also examined the transcriptomic landscape of CD4 T cells after stimulation conditions. Treatment with Tat-LNP did not change the transcriptomic landscape of the CD4 T cells overall, however, several transcripts were found to be differentially expressed between p24+ and p24- cells. These include protein coding genes enriched in cytotoxic T cells and pathways implicated in cell survival, proliferation and protein translation, and a long non-coding RNA.

Overall assessment of quality and importance

This is a well written and clearly presented manuscript describing well performed experiments that yielded several novel and interesting findings. The combination of Tat-LNP and PNB has several advantages over other latency-reversing treatments. Lipid nanoparticle-based delivery of Tat mRNA prevents uptake by endosomes, and produces low to no cellular toxicity and no effect on the cellular transcriptome at the bulk level. The findings imply a synergistic effect of exogenous Tat with an HDAC inhibitor in inducing HIV reactivation and virus particle release at a higher level than classic mitogens possibly due to targeting of different HIV transcription steps. Genes and pathways associated with persistent infected cells in people on long-term ART were identified. The heightened PI3K/Akt signaling, downregulation of protein translation and new findings regarding LINC02964 and SOD1P3 are interesting and deserve further exploration. The use of this latency

reversal approach shows promise both for future research studies and potentially as a therapy.

Major questions and comments:

- It is very interesting that many p24+ cells detected after Tat-LNP/PNB treatment contained proviruses with psi/MSD deletions. However, this finding raises several important points.
 - The Discussion does not address whether such proviruses could be part of the “rebound-competent” reservoir. This issue should be discussed explicitly.
 - Do the authors feel that Tat-LNP/PNB reactivation is skewed towards proviruses with psi/MSD deletions? Alternatively, are the low levels of intact proviruses in p24+ cells in this study simply what would be expected based on relative frequencies of psi/MSD-deleted vs. intact?
 - On page 12, the manuscript states that “(i) the proportions of intact and PSI/MSD defective proviruses were similar between Tat-LNP/PNB and PMA/i-reactivated p24+ cells.” This reviewer cannot find the data demonstrating this point. Please clarify. One might predict that MSD deletions would be best overcome by supplementation of Tat.
 - The manuscript should consider the low CD4 staining in p24+ cells more fully. If the proviruses in many of these cells have MSD deletions, how can they make Nef? Is it possible that functional Nef protein is encoded by messages that use alternative splice sites? Might it be possible to demonstrate alternative splice site usage that could lead to nef message using the RNAseq data?
- The manuscript does not address if latency reversal by Tat-LNP/PNB is sufficient to bring about death of infected cells. Can the authors comment on this?
- The microarray data demonstrate that Tat-LNP does not change the cellular transcriptome significantly, when considering a bulk cell population composed mainly of uninfected cells. However, it is not clear whether this necessarily means that genes and pathways enriched in p24+ cells reflect the state of these infected cells before stimulation. Is it possible that some of these genes and pathways were induced by the expression of the virus within the cell?
- This reviewer found Fig 5 confusing.
 - The legend or images for Fig 5 should more clearly indicate that panel A represents microarray data while B-D represent single-cell RNAseq.
 - It is confusing that there are yellow dots representing cells from unstimulated control on the 5B plot. These cells cluster with some of the Tat-LNP cells. What does this mean?
 - The PMA/i dots at the top-left of 5B and 5C are easy to miss.

Other points:

- Please add the number of p24+/p24- cells for each population in Fig 2B.
- Legend for colors used is missing 2C. Would suggest coloring bars same shade so as not to confuse with 2D.
- Do the dots connect the samples obtained from same patient in Fig 2? If yes, this should be mentioned in text. Would also be helpful to color the dots based on patients. This would make it easy to follow data from each patient across conditions.
- Would be helpful to know how many reads mapped to HXB2 when treated with PMA/i in Supplementary Fig 4.
- Correlation plot in Fig 5 - how is the correlation calculated? This reviewer would have expected correlation coefficients between -1 and 1. Please clarify the units.
- Title for Fig 5 and the results section pertaining to this figure should be revised. The current title seems to indicate results from only p24+ and p24- are reported.
- Heatmap shows that the clustering between p24+ and p24- cells is not perfect. This could indicate subpopulations within these two groups. Whether these are due to technical or biological variation should be addressed somewhere in the text.

Reviewer #3:

Remarks to the Author:

In this manuscript, Pardons et al describe a new method for reactivating latent HIV from CD4+ T cells that not induce major T cell activation. Specifically, they show that a nanoparticle containing Tat mRNA (Tat-LNP) in combination with the histone deacteylase inhibitor panobinostat induces HIV latency reversal in a high proportion of infected cells vs PMA/ionomycin. Using Tat-LNP +/- panobinostat, the authors identify transcriptomic differences between latently infected vs uninfected cells.

Overall, characterization of latently infected cells is an important area of research, and this study contributes toward the body of knowledge.

Comments:

1. Panobinostat is known to influence T cell activation (<https://journals.asm.org/doi/10.1128/mSphere.00616-17>) and perhaps the cell populations that express HIV-1 following PNB exposure (TCM/TTM) could be driven by this phenotype. The authors should more carefully measure T cell activation in the populations of cells they are using.
2. for the p24- controls in the scRNAseq analyses, it would be useful to know the frequency of HIV-1 infection (quantified by intracellular HIV DNA).
3. there should be a more detailed analysis and comparison of the genesets found here vs those recently published in other journals, particularly for candidate genes such as CCL5, IL7R etc. How much overlap is observed in candidate genes vs pathways.
4. Unclear how LINC02964 affects latency? Does it? or does it just suggest infection? More mechanistic studies of this lnc RNA would significantly strengthen the manuscript, given the focus on HIV latency and latency reversal.
5. The role of PI3K/Akt in latency reversal should be carefully assessed in light of published papers. Here, the authors show increase PI3K/Akt signaling - based on gene expression. Perhaps this should be validated by western blot/proteomics? It is well known that inhibition of PI3K blocks HIV latency reversal..

Reviewer #1 (Remarks to the Author):

In the manuscript from Pardons et al., entitled " Potent Latency Reversal by TAT RNA Enables Multi-Omic Analysis of the HIV-1 Reservoir" explored the HIV-1 reactivation capacity of a lipid nanoparticle containing Tat mRNA (Tat-LNP) in CD4 T cells from ART-treated individuals. The Authors found that the Tat-LNP, when combined with panobinostat, was able to induce HIV-1 latency reversal at a higher proportion of latently infected cells compared to PMA/ionomycin(≈4-fold).

Gene expression analysis showed that Tat-LNP alone does not significantly alter the transcriptome of CD4 T cells. The Authors identified transcriptomic differences between infected cells carrying an inducible provirus and non-infected cells, including the upregulation including GZMA, CCL5 and a long non-coding RNA LINC02964. Moreover, p24+ cells exhibit heightened PI3K/Akt signaling.

General comments

The Authors are showing not very novel as data as validation of their tat Mediated HIV-activation strategy has been already shown by others and even by the same Authors but sent to another. Besides the lack of novelty of their main finding, the remaining data presented in this manuscript is merely observational with no real understanding of the biology behind. Paradigmatic are the results obtained by scRNA sequencing, which were confined to very few cells, thus reducing the confidence of this reviewer on the solidity of their results.

The activation of a long non-coding RNA LINC02964 finding is potentially interesting, but no data at all on the functions is provided.

Altogether, I consider a low priority paper for Nature Communications.

We thank the reviewer for the assessment of our paper. The primary goal of the manuscript submitted by Van Gulck *et al* (doi: 10.1101/2023.03.02.530914) was to explore the reactivation capacity of a truncated Tat protein (Tat66), primarily in cell lines. While few data have been obtained with the mRNA formulation of Tat66 in the paper from Van Gulck *et al*, our manuscript focuses exclusively on this mRNA formulation and more specifically on its synergistic effect with Panobinostat. We would like to emphasize that there is no data overlap between the two manuscripts. Additionally, our paper highlights the potential of using Tat mRNA as a tool for studying viral and cellular transcripts expressed by translation-competent reservoir cells. Indeed, we successfully applied RNA sequencing through Smart-seq2 on single-sorted p24+ cells. Despite fixation and permeabilization of the cells, we detected between 1500 and 2000 genes per cell, which closely aligns with results from previous studies on non-fixed cells. Besides obtaining the transcriptome of the p24+ cells, we also retrieved complete coverage of the viral genome in around 50% of the p24+ cells, irrespective of the HIV subtype. This is a major achievement, as previous studies that attempted to assess the transcriptome of HIV-1 infected cells could only partially reconstruct the viral sequence in most of the cells (Cohn et al, Nat Med., 2018; Liu et al, Sci Transl Med., 2020), or could not assess it at all (Collora et al, Immunity, 2022; Clark et al, Nature, 2023).

Smart-seq2 processing yielded a total of 108 and 212 p24+ cells following treatment with Tat-LNP alone or combined with PNB, respectively. In parallel, a total of

109 and 150 p24⁻ cells were sorted after stimulation with Tat-LNP or Tat-LNP/PNB, respectively. The number of p24⁺ cells examined in our study directly mirrors the extremely low frequency of such cells. As a comparison point: (i) 85 Env⁺ cells and 109 Env⁻ cells from 10 ART-treated individuals were assessed in the LURE assay (Cohn et al, Nat Med., 2018); (b) 28 HIV RNA⁺ cells and 43 HIV RNA⁻ cells from 10 ART-treated individuals were assessed in the SortSeq assay (Liu et al, Sci Transl Med., 2020), (c) 9 HIV RNA⁺ cells from 6 ART-treated individuals were detected using the ECCITE-seq approach (Collora et al, Immunity, 2022). While we acknowledge that the number of p24⁺ cells we examined may be limited, it is essential to note that our single-cell RNA-seq study analyzed the highest number of translation-competent reservoir cells reported to date. Furthermore, we would like to emphasize that we validated several hits obtained by the single-cell RNA-seq approach (GZMA, CCL5, IL7R) at the protein level, using flow cytometry. In addition, pathway analysis by GSEA and module scoring gave results that were concordant across all participants, further reinforcing the reliability of our findings.

In the original version of our manuscript, we showed that the lncRNA *LINC02964* is highly upregulated in response to *in vitro* HIV infection. To complement these data, we conducted additional experiments that are presented in Fig 8. In p24⁺ cells isolated from viremic individuals, *LINC02964* expression was around 140 times higher than in the p24⁻ counterparts (Fig 8C), confirming our previous data obtained in *in vitro* infected CD4 T cells (Fig 8A-B). Furthermore, the knockdown of *LINC02964* using antisense oligonucleotides (ASOs) prior *in vitro* infection was associated with a reduced percentage of p24⁺ cells (Fig 8D-E), suggesting that *LINC02964* favors active HIV infection. Finally, to study the role of *LINC02964* in latency reversal, we developed a novel latent model (SupT1 cells harboring a latent single-round NL4.3-ΔENV-IRES-HSA): the knockdown of *LINC02964* in this model prior to Tat-LNP stimulation did not affect latency reversal, suggesting that *LINC02964* does not play a role in HIV reactivation (Fig 8F-G). While we agree that a better understanding of the function of this lncRNA is of high interest, we believe this is out of the scope of this study. Experiments aimed at understanding the precise function of *LINC02964* in HIV pathogenesis are currently ongoing, but still require an extensive amount of work.

To conclude, we would like to address the reviewer's comment regarding the perceived lack of novelty:

1. This manuscript presents the first combination of LRAs that reactivates a higher proportion of translation competent reservoir cells compared to classic mitogens. This formulation has potential for clinical translation, given that Panobinostat is FDA-approved for other indications. Furthermore, this combination of LRAs will benefit the research community by allowing to achieve high levels of HIV reactivation in an *in vitro* setting.

2. We are the first to study the transcriptome of reactivated translation-competent reservoir cells using a compound that does not alter the transcriptome of CD4 T cells.

3. We report several novel single-cell RNA-seq hits, including *LINC02964*, *SOD1P3*, along with the observation that p24⁺ cells display a lower level of translation compared to p24⁻ cells.

Reviewer #2 (Remarks to the Author):

Summary of findings:

Pardons et. al. examined the reactivation potential of Tat mRNA encapsulated in a lipid nanoparticle (Tat-LNP) and used it to study the properties of HIV-infected CD4 T cells from ART treated patients. In combination with an HDAC inhibitor Panobinostat (PNB), Tat-LNP was found to be more effective in reversing latency but less cytotoxic compared with PMA/ionomycin (PMA/i). After stimulation with Tat-LNP/PNB significant enrichment of p24+ cells were found in the effector memory population compared to the Naïve or terminally differentiated cells.

Integration site as well proviral sequences between different conditions from five participants on ART were identified. Overall, proviral landscapes were similar between Tat-LNP/PNB and PMA/i conditions. Higher degree of latency reversal was noted after treating with Tat-LNP/PNB than with Tat-LNP alone as shown by the HXB2 mapped reads. The study also examined the transcriptomic landscape of CD4 T cells after stimulation conditions. Treatment with Tat-LNP did not change the transcriptomic landscape of the CD4 T cells overall, however, several transcripts were found to be differentially expressed between p24+ and p24- cells. These include protein coding genes enriched in cytotoxic T cells and pathways implicated in cell survival, proliferation and protein translation, and a long non-coding RNA.

Overall assessment of quality and importance

This is a well written and clearly presented manuscript describing well performed experiments that yielded several novel and interesting findings. The combination of Tat-LNP and PNB has several advantages over other latency-reversing treatments. Lipid nanoparticle-based delivery of Tat mRNA prevents uptake by endosomes, and produces low to no cellular toxicity and no effect on the cellular transcriptome at the bulk level. The findings imply a synergistic effect of exogenous Tat with an HDAC inhibitor in inducing HIV reactivation and virus particle release at a higher level than classic mitogens possibly due to targeting of different HIV transcription steps. Genes and pathways associated with persistent infected cells in people on long-term ART were identified. The heightened PI3K/Akt signaling, downregulation of protein translation and new findings regarding LINC02964 and SOD1P3 are interesting and deserve further exploration. The use of this latency reversal approach shows promise both for future research studies and potentially as a therapy.

We thank the reviewer for the careful assessment of our manuscript and the pertinent questions raised about our work.

Major questions and comments:

- It is very interesting that many p24+ cells detected after Tat-LNP/PNB treatment contained proviruses with psi/MSD deletions. However, this finding raises several important points.

- o The Discussion does not address whether such proviruses could be part of the “rebound-competent” reservoir. This issue should be discussed explicitly.

Recent evidence indicates that proviruses with PSI/MSD defects exhibit limited infectivity due to a reduced production of 4kb transcripts, which encode for the envelope protein. While these observations suggest that PSI/MSD-defective proviruses are not replication-competent and would therefore not contribute to viral rebound following ART cessation, those proviruses can be the source of non-suppressible viremia (NSV) in ART-treated participants (White et al, J clin Invest, 2023). Therefore, the detailed characterization of reservoir cells carrying PSI/MSD-defective proviruses is of high interest, especially in the context of participants experiencing NSV. Future studies involving cloning of full-length HIV sequences with various PSI/MSD defects are required to completely rule out the potential implication of PSI/MSD-defective proviruses in viral rebound. This is now explicitly discussed at lines 361-367.

- o Do the authors feel that Tat-LNP/PNB reactivation is skewed towards proviruses with psi/MSD deletions? Alternatively, are the low levels of intact proviruses in p24+ cells in this study simply what would be expected based on relative frequencies of psi/MSD-deleted vs. intact? On page 12, the manuscript states that “(i) the proportions of intact and PSI/MSD defective proviruses were similar between Tat-LNP/PNB and PMA/i-reactivated p24+ cells.” This reviewer cannot find the data demonstrating this point. Please clarify. One might predict that MSD deletions would be best overcome by supplementation of Tat.

As mentioned on page 12, the proportions of intact and PSI/MSD-defective proviruses are similar between Tat-LNP/PNB and PMA/i-reactivated p24+ cells. Therefore, Tat-LNP/PNB reactivation does not appear to skew the proportions of intact vs PSI/MSD-defective proviruses in p24+ cells. We apologize for the absence of a figure showing those results. We now added a figure (Figure 3A) showing the proportions of intact vs PSI/MSD-defective proviruses in the PMA/i and Tat-LNP/PNB conditions. A table containing the raw counts of intact vs PSI/MSD-defective proviruses retrieved from p24+ cells in each participant is also shown besides the Figure. A detailed information about each individual provirus retrieved from p24+ cells can be found in Supplementary Data 1.

- o The manuscript should consider the low CD4 staining in p24+ cells more fully. If the proviruses in many of these cells have MSD deletions, how can they make Nef? Is it possible that functional Nef protein is encoded by messages that use alternative splice sites? Might it be possible to demonstrate alternative splice site usage that could lead to nef message using the RNAseq data?

Several studies have shown that PSI/MSD-defective proviruses can use alternative splice sites to produce spliced transcripts (Pollack et al, Cell Host Microbe, 2017; White et al, J clin Invest, 2023). These findings support that PSI/MSD-defective proviruses are capable

of producing Nef, enabling CD4 downregulation. This is now addressed in the manuscript at lines 358-361.

Using our single-cell RNA-seq data on p24+ cells, we confirmed that PSI/MSD-defective proviruses can use alternative splice sites located downstream or upstream of the PSI/MSD defect. These data are represented on the figure below, for the reviewer's consideration. We are currently performing long-read sequencing experiments to further assess the type of transcripts produced by PSI/MSD-defective proviruses, which we believe falls out of the scope of the current manuscript and will be the subject of a separate manuscript.

Single-cell RNA-seq allows for the detection of splice donor sites in p24+ cells. A) Virogram depicting the 5' UTR region of the viral sequences retrieved with Smart-seq2. Viral sequences are grouped into 3 categories: intact MSD (D1) and CD (D1c), defective MSD but intact CD, defective MSD and CD. Type of defects, splice site usage and participants are color-coded. Clonal sequences are represented only once in the virogram. HXB2 is used as a reference genome. B) Summary of the deletions (top) and splice sites (bottom) detected over the entire dataset. D1 = major splice donor, D1c = cryptic donor.

- The manuscript does not address if latency reversal by Tat-LNP/PNB is sufficient to bring about death of infected cells. Can the authors comment on this?

We thank the reviewer for raising this interesting point. Given the inherent challenges associated with employing primary CD4 T cells from ART-treated individuals to address this specific question, we turned to the Jurkat/J-Lat10.6 cell line model. Cells were stimulated overnight with Tat-LNP alone (1.4nM, 0.35nM), PNB alone (50nM, 12.5nM)

and the combination Tat-LNP/PNB (1.4nM/50nM, 0.35nM/12.5nM). Viability of the cells was assessed by propidium iodide (PI) staining using flow cytometry (left panel), or using the Cell Titer Glo luminescent assay (Promega) which quantifies ATP as a surrogate marker of metabolically active cells (middle panel)(n=2-3 independent experiments). In parallel, the frequency of GFP+ cells was measured by flow cytometry to assess HIV reactivation (right panel).

At both concentrations tested, Tat-LNP did not appear to display an increased toxicity in the J-Lat10.6 cell line compared to the parental Jurkat cell line (left and middle panel), despite inducing high levels of reactivation (means = 69.8% and 66.2% GFP+ cells at 1.4nM and 0.35nM, respectively)(right panel). In contrast, when cells were stimulated with PNB or Tat-LNP/PNB, lower percentages of viable cells (PI- cells) were observed in the J-Lat10.6 cell line relative to the Jurkat cell line, suggesting that PNB may sensitize HIV-infected cells to cell death. However, this observation was somewhat less pronounced in the Cell Titer Glo luminescent assay. While these findings suggest that Tat-LNP-induced reactivation may not trigger cell death through the cytopathic effects of the virus, we can not exclude that killing of HIV-infected cells is induced by the immune system following reactivation. Indeed, SIMOA experiments show that Tat-LNP and Tat-LNP/PNB-induced reactivation is associated with the release of viral particles into the supernatant (Figure 1C), suggesting that the reactivation signal is potent enough to allow for the killing of infected cells by cytotoxic CD8 T cells. Future experiments conducted *ex vivo* on primary cells or *in vivo* in humanized mice will help shed further light on this important question.

- The microarray data demonstrate that Tat-LNP does not change the cellular transcriptome significantly, when considering a bulk cell population composed mainly of uninfected cells. However, it is not clear whether this necessarily means that genes and pathways enriched in p24+ cells reflect the state of these infected cells before stimulation. Is it possible that some of these genes and pathways were induced by the expression of the virus within the cell?

The reviewer is correct: our setup does not allow to define whether the identified markers and pathways upregulated by p24+ cells are the result of HIV reactivation itself or whether they reflect the initial state of the latently infected cells before stimulation. We touch upon this topic in the discussion: “Further investigations are needed to determine whether these signatures are caused by HIV reactivation or if they reflect unique cellular traits that confer long-term survival benefits”, lines 395-396.

- This reviewer found Fig 5 confusing.
- o The legend or images for Fig 5 should more clearly indicate that panel A represents microarray data while B-D represent single-cell RNAseq.

We agree with the reviewer that Fig 5 is confusing. To make it clearer, we decided to only keep single-cell RNA sequencing data in the main figure and to display microarray-related figures on Supplementary Fig 5.

o It is confusing that there are yellow dots representing cells from unstimulated control on the 5B plot. These cells cluster with some of the Tat-LNP cells. What does this mean? A total of 17 NS cells (yellow dots) were sorted. NS cells indeed cluster with Tat-LNP-stimulated cells, confirming that Tat-LNP does not significantly impact the transcriptome of CD4 T cells. This is now better specified in the manuscript, at lines 245-247: **“Tat-LNP-stimulated cells clustered with the NS cells, further supporting that Tat-LNP does not modify the transcriptome of CD4 T cells, while PMA/i and Tat-LNP/PNB-treated cells each formed a separate cluster (Fig 5A).**

o The PMA/i dots at the top-left of 5B and 5C are easy to miss. A total of 28 PMA/i-stimulated p24+ cells (green dots) were sorted. In order to not miss the PMA/i dots at the top-left, we added ellipses to define the 3 clusters of cells.

Other points:

- Please add the number of p24+/p24- cells for each population in Fig 2B. The number of p24+ and p24- cells were added on Fig 2B.
- Legend for colors used is missing 2C. Would suggest coloring bars same shade so as not to confuse with 2D. As suggested by the reviewer, we colored the bars in the same shade, both in Fig 2C and Fig 2D. The legend (p24- cells vs p24+ cells for Fig 2C, PMA/i vs Tat-LNP/PNB for Fig 2D) is now indicated below the bars.
- Do the dots connect the samples obtained from same patient in Fig 2? If yes, this should be mentioned in text. Would also be helpful to color the dots based on patients. This would make it easy to follow data from each patient across conditions. Indeed, the dotted lines connect results obtained from the same participant. Individuals that were used to generate both Fig 2C and Fig 2D are now color-coded, as suggested by the reviewer. Of note, the colors used for each participant are kept throughout the entire manuscript. Data from participants that were only used for Fig 2C (and not for Fig 2D nor for the rest of the paper) are represented in dark grey. While Fig 2C was generated using only HIV-Flow data (PFA-fixed cells), Fig 2D combined data obtained from HIV-Flow and STIP-seq experiments (PFA-fixed and methanol-fixed cells) in order to increase the number of p24+ cells analyzed in PMA/i and Tat-LNP/PNB conditions. This information is now added to the legend of the figure.
- Would be helpful to know how many reads mapped to HXB2 when treated with PMA/i in Supplementary Fig 4. This figure is now modified: the proportion of reads mapping to HXB2 in PMA/i-stimulated p24+ cells has been added to the graph. The proportion of reads mapping to HXB2 is lower

in the PMA/i condition compared to Tat-LNP and Tat-LNP/PNB-stimulated cells (medians = 5.4%, 6.7% and 15% for PMA/i, Tat-LNP alone and Tat-LNP/PNB, respectively). This is likely the result of several non-mutually exclusive factors: (i) PMA/i-stimulated p24+ cells were only obtained from two participants (MRC03 and MRC15; n = 28), (ii) 15/16 p24+ cells from MRC15 belong to the same clone, (iii) 6/12 p24+ cells from MRC03 display a TN phenotype, which may be associated with a lower expression of HIV transcripts, (iv) PMA/i induces a significant upregulation of a high number of genes, thus diluting HIV transcripts.

- Correlation plot in Fig 5 - how is the correlation calculated? This reviewer would have expected correlation coefficients between -1 and 1. Please clarify the units.

Fig 5A (now Supplementary Fig 5A) shows distance (dis)-similarity matrix (with value range from 0 – 170), where similar samples are grouped together (based on small distance measure) and are color coded as well as depicted by the size of the dendrogram.

We computed the distance matrix using a Euclidean distance measure to calculate the distances between the rows of a data matrix. In 2-dimensional space, the Euclidean distance between two points (Point1 and Point2) is calculated from the cartesian coordinates of the points (x1, y1) and (x2, y2). The Euclidean distance is given by: $\sqrt{(x1 - x2)^2 + (y1 - y2)^2}$. This is now specified in the Material and methods section, at lines 515-517.

For example, the Euclidean distance between two vectors:

- Sample 1 is the first vector: (8, 4, 6, 8)
- Sample 2 is the second vector: (1, 5, 8, 2)

Their Euclidean distance is calculated as, $\sqrt{(8 - 1)^2 + (4 - 5)^2 + (6 - 8)^2 + (8 - 2)^2}$ which is equal to 9.46.

A 28 x 28 data matrix containing the full Euclidean measure for all samples in the microarray experiment is now provided in the source data file.

- Title for Fig 5 and the results section pertaining to this figure should be revised. The current title seems to indicate results from only p24+ and p24- are reported.

As we decided to only keep single-cell RNA-seq data in the main figure (Fig 5) and to have all microarray-related figures in Supplementary Fig 5, the title of Fig 5 seems to be appropriate. Regarding the results section, the paragraph containing the microarray and the Smart-seq2 data was split in two, with the following titles: “Tat-LNP does not modify the transcriptome of CD4 T cells” (microarray data; line 217), and “P24+ cells display a distinct transcriptional landscape compared to p24- cells” (Smart-seq2 data; line 236).

- Heatmap shows that the clustering between p24+ and p24- cells is not perfect. This could indicate subpopulations within these two groups. Whether these are due to technical or biological variation should be addressed somewhere in the text.

We agree with the reviewer that the unsupervised clustering does not perfectly segregate p24+ from p24- cells, which is in agreement with the relatively small number of DEG we retrieved (n = 82) and the similar nature of those two populations (p24+ and p24- cells are CD45RO- memory CD4 T cells). This is now specified in the text at lines 259-261. In line with those data, previous findings obtained with the LURE assay (Cohn et al, Nat

Med., 2018) show that gag+Env+ cells do not perfectly segregate from their non-infected counterparts (see figure below), again supporting the notion that those two populations are highly alike and differ only in a small set of genes.

For each participant, p24- and p24+ cells were sorted on the same day. Furthermore, Smart-seq2, library preparation and sequencing were also performed on the same day for the two populations of cells. Therefore, we do not believe that our findings are impacted by technical variations. This is now specified in the text at lines 241-243.

Reviewer #3 (Remarks to the Author):

In this manuscript, Pardons et al describe a new method for reactivating latent HIV from CD4+ T cells that not induce major T cell activation. Specifically, they show that a nanoparticle containing Tat mRNA (Tat-LNP) in combination with the histone deacetylase inhibitor panobinostat induces HIV latency reversal in a high proportion of infected cells vs PMA/ionomycin. Using Tat-LNP +/- panobinostat, the authors identify transcriptomic differences between latently infected vs uninfected cells.

Overall, characterization of latently infected cells is an important area of research, and this study contributes toward the body of knowledge.

Comments:

1. Panobinostat is known to influence T cell activation (<https://journals.asm.org/doi/10.1128/mSphere.00616-17>) and perhaps the cell populations that express HIV-1 following PNB exposure (TCM/TTM) could be driven by this phenotype. The authors should more carefully measure T cell activation in the populations of cells they are using.

We thank the reviewer for raising this interesting point. Several papers have indeed documented T cell activation in response to PNB stimulation: Kazmierski et al, *Biorxiv* 2023 (10.1101/2020.05.04.075119); Brinkmann et al, *mSphere* 2018 (10.1128/msphere.00616-17); Rasmussen et al, *Hum Vaccin Immunother.* 2013 (10.4161/hv.23800). To confirm those results, we stimulated CD4 T cells from four individuals (2 HIV negative, 2 ART-treated individuals) with PNB 50nM for 24h, and assessed the expression of activation markers (CD69, CD25 and HLA-DR) by flow cytometry. In the CD4 T cell fraction, the mean fluorescence intensity (MFI) of CD69 greatly increased following PNB stimulation compared to the non stimulated condition (mean MFI = 661.3 and 86.0, respectively), while the MFI of CD25 and HLA-DR remained unaltered. This figure was added to the manuscript (Supplementary Fig 2B). We then assessed the expression of those activation markers in the CD4 T cell subsets (TN, TCM, TEM, TTd) (Supplementary Fig 2D). In the absence of stimulation, CD69 was expressed at higher levels in more differentiated subsets (mean MFI in TN: 73.3 < TCM: 99.8 < TEM: 131.8 < TTd: 176.8). Following PNB stimulation, the highest levels of CD69 expression were observed in the TCM fraction (mean MFI in TCM: 853.5 > TEM: 731.0 > TN: 594.3 > TTd: 499.3). Moreover, the fold induction over the NS condition decreased progressively with T cell differentiation (mean fold induction in TN: 12.8 > TCM: 9.0 > TEM: 5.5 > TTd: 2.8) (Supplementary Fig 2E). The increased effectiveness of PNB in promoting CD69 expression within the TCM subset compared to the TEM subset, could offer a partial explanation for the observed tendency toward a greater presence of p24+ cells with a TCM phenotype following Tat-LNP/PNB stimulation.

2. for the p24- controls in the scRNAseq analyses, it would be useful to know the frequency of HIV-1 infection (quantified by intracellular HIV DNA).

Total HIV DNA was not measured specifically in the p24- fraction. Nevertheless, we did measure total HIV DNA by digital PCR in bulk CD4 T cells. The table below displays the frequency of total HIV DNA copies per million CD4 T cells, the frequency of p24+ cells per million CD4 T cells following Tat-LNP and Tat-LNP/PNB stimulation, as well as the fraction of the total HIV-1 reservoir which is reactivated in response to stimulation.

Participant ID	Total HIV DNA (cps/10 ⁶ CD4 T cells)	Frequency of p24+ cells (per 10 ⁶ CD4 T cells)		Frequency of p24+ cells/ total HIV DNA (%)	
		Tat-LNP	Tat-LNP/PNB	Tat-LNP	Tat-LNP/PNB
MRC01	2592.3	2.0	12.7	0.1	0.5
MRC04	779.8	2.9	10.5	0.4	1.3
MRC08	450.5	28.5	15.7	6.3	3.5
MRC15	183.6	33.3	52.4	18.1	28.5
MRC21	5690.9	6.3	33.0	0.1	0.6

This information can be found in Supplementary Table 2.

3. there should be a more detailed analysis and comparison of the genesets found here vs those recently published in other journals, particularly for candidate genes such as CCL5, IL7R etc. How much overlap is observed in candidate genes vs pathways.

To compare our findings to those reported in other publications, we now include an overview table summarizing the most important findings of recent studies (Supplementary Table 3). Apart from the most important genes and pathways reported, this table also includes information on the technology used to generate the data, enrichment strategies, etc.

To illustrate the connection between the identified DEGs (adjusted p value < 0.05; n = 82) (Fig 5, Supplementary Data 5) and the genes within the enriched gene sets (Fig 6, Supplementary Data 6), we include here, for the reviewer's consideration, a Venn diagram showing the overlap between the identified DEGs and the genes comprised in the enriched gene sets. Significantly, this reveals that the DEG CCL5 is a component of the GOBP_POSITIVE_REGULATION_OF_PI3K_SIGNALING gene set. Furthermore, 9 DEGs identified in our study (ribosomal genes) are comprised in the GOBP_CYTOPLASMIC_TRANSLATION gene set. If you require information on the primary genes driving the enrichment for each gene set, you can refer to the leading edge column in Supplementary Data 6.

4. Unclear how LINC02964 affects latency? Does it? or does it just suggest infection? More mechanistic studies of this lnc RNA would significantly strengthen the manuscript, given the focus on HIV latency and latency reversal.

As mentioned in our response to reviewer #1, we conducted additional experiments that are presented in Fig 8. We showed that, in p24+ cells isolated from three viremic individuals, LINC02964 expression is around 140X higher than in the p24- counterparts (Fig 8C), confirming our previous data obtained in *in vitro* infected CD4 T cells (Fig 8A-B).

Furthermore, the knockdown of *LINC02964* using antisense oligonucleotides (ASOs) prior *in vitro* infection was associated with a reduced percentage of p24+ cells (Fig 8D-E), suggesting that *LINC02964* favors active HIV infection. Finally, to answer the question of the reviewer about the role of *LINC02964* in latency reversal, we developed a novel latent model (SupT1 cells harboring a latent single-round NL4.3- Δ ENV-IRES-HSA): the knockdown of *LINC02964* in this model prior to Tat-LNP stimulation did not affect latency reversal, suggesting that *LINC02964* does not play a role in HIV reactivation (Fig 8F-G).

5. The role of PI3K/Akt in latency reversal should be carefully assessed in light of published papers. Here, the authors show increased PI3K/Akt signaling - based on gene expression. Perhaps this should be validated by western blot/proteomics? It is well known that inhibition of PI3K blocks HIV latency reversal.

We appreciate the reviewer's insightful suggestion and agree with the significance of validating the pathways highlighted in this study. Due to the low number of p24+ cells retrieved in each participant, confirming the heightened PI3K/Akt pathway in p24+ cells through Western Blot would not be feasible. Therefore, we decided to turn to the latently infected J-Lat 10.6 cell line. We employed a Western Blot protocol targeting phospho-Akt (pAkt, Ser473; CST #4060) and total Akt (recognizing Akt1, 2, and 3; CST #4691), both with and without Tat-LNP stimulation. Complete stripping between the two staining steps was reached using the Restore™ PLUS Western Blot Stripping Buffer (Thermo Scientific #46430).

Notably, the percentage of GFP+ cells increased from 4% in the absence of stimulation to 75% post Tat-LNP stimulation (Figure A). Due to the high percentage of GFP+ cells following Tat-LNP stimulation, no sorting strategy was employed to enrich for reactivated cells. While our Western Blot protocol effectively identified pAkt (Figure B) and Total Akt (Figure C), we encountered challenges due to elevated pAkt background levels in unstimulated J-Lat10.6 cells, hindering our ability to observe a discernible rise in pAkt signaling post Tat-LNP stimulation.

Further validation would require to apply this Western Blot protocol in the context of a primary cell model of HIV latency, or to develop a flow cytometry protocol allowing to assess pAkt expression in p24+ vs p24- cells. Nevertheless, we believe this falls beyond the scope of this manuscript.

Reviewers' Comments:

Reviewer #1:

Remarks to the Author:

The Authors have produced additional data and have significantly improved the overall manuscript. revised version of the manuscript is satisfactory, and I have no additional concerns or questions.

Reviewer #2:

Remarks to the Author:

The revised manuscript and rebuttal letter by Pardons et. al. address the concerns raised by the initial version of the manuscript. New data and textual revisions are very helpful. This reviewer especially appreciated the new experiments regarding the role of LINC02964 in HIV infection and reactivation. The amount of additional work performed is commendable and has resulted in a stronger story.

Below, minor comments in response to selected rebuttal statements provided by the authors are indicated with ">" symbols. Additional minor suggestions are included at the end, also indicated by ">" symbols.

Original review: The manuscript should consider the low CD4 staining in p24+ cells more fully. If the proviruses in many of these cells have MSD deletions, how can they make Nef? Is it possible that functional Nef protein is encoded by messages that use alternative splice sites? Might it be possible to demonstrate alternative splice site usage that could lead to nef message using the RNAseq data?

Rebuttal: Several studies have shown that PSI/MSD-defective proviruses can use alternative splice sites to produce spliced transcripts (Pollack et al, Cell Host Microbe, 2017; White et al, J clin Invest, 2023). These findings support that PSI/MSD-defective proviruses are capable of producing Nef, enabling CD4 downregulation. This is now addressed in the manuscript at lines 358-361.

Using our single-cell RNA-seq data on p24+ cells, we confirmed that PSI/MSD-defective proviruses can use alternative splice sites located downstream or upstream of the PSI/MSD defect. These data are represented on the figure below for the reviewer's consideration. We are currently performing long-read sequencing experiments to further assess the type of transcripts produced by PSI/MSD-defective proviruses, which we believe falls out of the scope of the current manuscript and will be the subject of a separate manuscript.

>In this reviewer's opinion, many readers will question how viruses with MSD defects can drive CD4 downregulation. Although the cited paper by Pollack et al. does address the issue somewhat, directly demonstrating alternatively spliced transcripts in the CD4-low cells would be stronger. The authors should consider including the data presented in the rebuttal letter figure in the final manuscript. The most effective display item might be a read mapping demonstrating the alternative splice junctions directly.

Original review: Heatmap shows that the clustering between p24+ and p24- cells is not perfect. This could indicate subpopulations within these two groups. Whether these are due to technical or biological variation should be addressed somewhere in the text.

Rebuttal: We agree with the reviewer that the unsupervised clustering does not perfectly segregate p24+ from p24- cells, which is in agreement with the relatively small number of DEG we retrieved (n = 82) and the similar nature of those two populations (p24+ and p24- cells are CD45RO- memory CD4 T cells). This is now specified in the text at lines 259-261. In line with those data, previous findings obtained with the LURE assay (Cohn et al, Nat 9 Med., 2018) show that gag+Env+ cells do not perfectly segregate from their non-infected counterparts (see figure below), again supporting the notion that those two populations are highly alike and differ only in a

small set of genes.

For each participant, p24- and p24+ cells were sorted on the same day. Furthermore, Smart-seq2, library preparation and sequencing were also performed on the same day for the two populations of cells. Therefore, we do not believe that our findings are impacted by technical variations. This is now specified in the text at lines 241-243.

>The provided explanation and the revised text are clear and helpful for understanding. The authors mention doing QC in their methods section. Were the samples similar in terms of their QC metrics, mapping % etc.? Would recommend adding this information to the manuscript.

Other minor comments:

>Supplementary Fig 2A doesn't have a pre-treatment/baseline and the measurements starts from 24 hr post-treatment. Would be helpful to know if there's any change between the pre-treatment/post-treatment time points.

>In the figures with dot plots and bar chart (e.g. Fig2C), some bars show the median values while others show mean. This can get confusing. Would recommend keeping things consistent across figures.

>In the microarray data examining effect of Tat-LNP on the transcriptome, a 6hr treatment is done. In other experiments, the treatment is generally for a longer period of time (24hrs or more). The authors should consider explaining why a shorter treatment time was used.

>In figure 7, different measurements for assessing different targets are used (e.g. %GZMA+ cells, IL7R MFI). The authors should consider providing an explanation for this.

>In Figure 5C, what is the statistical test used?

>For the differentially expressed genes, it would be helpful to know the fold changes cut-offs (if used) in addition to the adjusted p-values (e.g. lines 228, 231)

Reviewer #3:

Remarks to the Author:

The authors were extremely responsive to prior reviews and the quality of the study has been improved.

The revised manuscript and rebuttal letter by Pardons et. al. address the concerns raised by the initial version of the manuscript. New data and textual revisions are very helpful. This reviewer especially appreciated the new experiments regarding the role of LINC02964 in HIV infection and reactivation. The amount of additional work performed is commendable and has resulted in a stronger story.

Below, minor comments in response to selected rebuttal statements provided by the authors are indicated with ">" symbols. Additional minor suggestions are included at the end, also indicated by ">" symbols.

- 1) The manuscript should consider the low CD4 staining in p24+ cells more fully. If the proviruses in many of these cells have MSD deletions, how can they make Nef? Is it possible that functional Nef protein is encoded by messages that use alternative splice sites? Might it be possible to demonstrate alternative splice site usage that could lead to nef message using the RNAseq data?

Several studies have shown that PSI/MSD-defective proviruses can use alternative splice sites to produce spliced transcripts (Pollack et al, Cell Host Microbe, 2017; White et al, J clin Invest, 2023). These findings support that PSI/MSD-defective proviruses are capable of producing Nef, enabling CD4 downregulation. This is now addressed in the manuscript at lines 358-361.

Using our single-cell RNA-seq data on p24+ cells, we confirmed that PSI/MSD-defective proviruses can use alternative splice sites located downstream or upstream of the PSI/MSD defect. These data are represented on the figure below for the reviewer's consideration. We are currently performing long-read sequencing experiments to further assess the type of transcripts produced by PSI/MSD-defective proviruses, which we believe falls out of the scope of the current manuscript and will be the subject of a separate manuscript.

>In this reviewer's opinion, many readers will question how viruses with MSD defects can drive CD4 downregulation. Although the cited paper by Pollack et al. does address the issue somewhat, directly demonstrating alternatively spliced transcripts in the CD4-low cells would be stronger. The authors should consider including the data presented in the rebuttal letter figure in the final manuscript. The most effective display item might be a read mapping demonstrating the alternative splice junctions directly.

As requested by the reviewer, we included the data presented in the rebuttal letter in the final manuscript (Fig.4C). Usage of alternative splice sites by MSD-defective proviruses is now clarified in the text at lines 223-228.

- 2) Heatmap shows that the clustering between p24+ and p24- cells is not perfect. This could indicate subpopulations within these two groups. Whether these are due to technical or biological variation should be addressed somewhere in the text.

We agree with the reviewer that the unsupervised clustering does not perfectly segregate p24+ from p24- cells, which is in agreement with the relatively small number of DEG we retrieved (n = 82) and the similar nature of those two populations (p24+ and p24- cells are CD45RO- memory CD4 T cells). This is now specified in the text at lines 259-261. In line with those data, previous findings obtained with the LURE assay (Cohn et al, Nat 9 Med., 2018) show that gag+Env+ cells do not perfectly segregate from their non-infected counterparts (see figure below), again supporting the notion that those two populations are highly alike and differ only in a small set of genes.

For each participant, p24- and p24+ cells were sorted on the same day. Furthermore, Smart-seq2, library preparation and sequencing were also performed on the same day for the two populations of

cells. Therefore, we do not believe that our findings are impacted by technical variations. This is now specified in the text at lines 241-243.

>The provided explanation and the revised text are clear and helpful for understanding. The authors mention doing QC in their methods section. Were the samples similar in terms of their QC metrics, mapping % etc.? Would recommend adding this information to the manuscript.

To the reviewer's request, we have included a supplementary table displaying further information on the QC metrics: number of genes per cell, number of mapped reads per cell, and the percentage of mitochondrial reads per cell (Supplementary Table 5). Medians, interquartile ranges (IQR), minima and maxima are given for all single-sorted cells (all), single-sorted p24+ cells, and single-sorted p24- cells.

Other minor comments:

>Supplementary Fig 2A doesn't have a pre-treatment/baseline and the measurements starts from 24 hr post-treatment. Would be helpful to know if there's any change between the pre-treatment/post-treatment time points.

Flow cytometry stainings were not performed pre-treatment, which does not allow to answer the reviewer's question.

>In the figures with dot plots and bar chart (e.g. Fig2C), some bars show the median values while others show mean. This can get confusing. Would recommend keeping things consistent across figures.

To comply with the reviewer's request, we are now showing the median on all figures (at the exception of Fig.8 and Supplementary Fig.2B, D and E, which display a maximum of 4 datapoints).

>In the microarray data examining effect of Tat-LNP on the transcriptome, a 6hr treatment is done. In other experiments, the treatment is generally for a longer period of time (24hrs or more). The authors should consider explaining why a shorter treatment time was used.

This is now clarified in the text at lines 223-224, 247-248: "In order to capture the immediate cellular responses and initial signaling events to the compounds of interest, cells were stimulated for a period of 6h". "Prolonged exposure of the cells (24h) to the compounds of interest produced similar results (Supplementary data 4)".

>In figure 7, different measurements for assessing different targets are used (e.g. %GZMA+ cells, IL7R MFI). The authors should consider providing an explanation for this.

This is now clarified in the legend of Fig.7: "When a clear distinction between positive and negative subsets could be defined (GZMA, GZMB), results are expressed as a percentage of GZMA+/GZMB+ cells in the p24-/p24+ fractions; when a continuum of expression was observed with no clear distinction between positive and negative subsets (IL7R, CCL5), the results are expressed as IL7R/CCL5 MFI in the p24-/p24+ fractions (gating strategy in Supplementary Fig.7A)".

>In Figure 5C, what is the statistical test used?

The p-values from Fig.5C were derived from likelihood ratio tests (used in the MAST algorithm). P-values lower than 0.05 after Bonferroni corrections were considered significant. This is now clarified in the legend of the figure and in the Methods section.

>For the differentially expressed genes, it would be helpful to know the fold changes cut-offs (if used) in addition to the adjusted p-values (e.g. lines 228, 231)

We considered DEG with an adjusted p-value <0.05. We did not perform any additional cut-offs based on fold change.